# Repression of germline genes by PRC1.6 and SETDB1 in the early embryo precedes DNA methylation-mediated silencing

Kentaro Mochizuki [1], Jafar Sharif[2], Kenjiro Shirane [1,5], Kousuke Uranishi[3], Aaron B. Bogutz[1], Sanne M. Janssen [1], Ayumu Suzuki[3], Akihiko Okuda [3], Haruhiko Koseki [2,4] & Matthew C. Lorincz [1✉]

Silencing of a subset of germline genes is dependent upon DNA methylation (DNAme) post-implantation. However, these genes are generally hypomethylated in the blastocyst, implicating alternative repressive pathways before implantation. Indeed, in embryonic stem cells (ESCs), an overlapping set of genes, including germline "genome-defence" (GGD) genes, are upregulated following deletion of the H3K9 methyltransferase SETDB1 or subunits of the non-canonical PRC1 complex PRC1.6. Here, we show that in pre-implantation embryos and naïve ESCs (nESCs), hypomethylated promoters of germline genes bound by the PRC1.6 DNA-binding subunits MGA/MAX/E2F6 are enriched for RING1B-dependent H2AK119ub1 and H3K9me3. Accordingly, repression of these genes in nESCs shows a greater dependence on PRC1.6 than DNAme. In contrast, GGD genes are hypermethylated in epiblast-like cells (EpiLCs) and their silencing is dependent upon SETDB1, PRC1.6/RING1B and DNAme, with H3K9me3 and DNAme establishment dependent upon MGA binding. Thus, GGD genes are initially repressed by PRC1.6, with DNAme subsequently engaged in post-implantation embryos.

[1] Department of Medical Genetics, Life Sciences Institute, University of British Columbia, Vancouver, British Columbia, Canada. [2] Laboratory for Developmental Genetics, RIKEN Center for Integrative Medical Sciences (IMS), Yokohama, Kanagawa, Japan. [3] Division of Biomedical Sciences, Research Center for Genomic Medicine, Saitama Medical University, Hidaka, Saitama, Japan. [4] Department of Cellular and Molecular Medicine, Graduate School of Medicine, Chiba University, 1-8-1 Inohana, Chuo ward, Chiba, Japan. [5] Present address: Department of Stem Cell Biology and Medicine, Graduate School of Medical Sciences, Kyushu University, Higashi-ku, Fukuoka, Japan. ✉email: mlorincz@mail.ubc.ca

DNA methylation (DNAme) plays an essential role in transcriptional regulation in development and disease[1]. This epigenetic mark is highly dynamic during embryonic development, with a genome-wide wave of demethylation occurring after fertilization, mediated at least in part by dioxygenases of the ten-eleven translocation (TET) protein family[2]. Subsequently, a wave of global de novo DNAme occurs during post-implantation epiblast formation, mediated by DNMT3A and DNMT3B[3,4]. While DNAme is generally maintained at high levels in somatic lineages[5], another wave of demethylation occurs early in germ cell development in both males and females[6]. During this period, many genes, including those essential for meiosis and gametogenesis, are upregulated. Strict temporal regulation of such germline genes is clearly critical for normal development, as their ectopic expression in somatic cells can contribute to tumorigenesis[7,8]. Furthermore, premature activation of germline genes in early germ cells, and in turn precocious induction of meiosis/gametogenesis pathways, leads to reduced germ cell numbers and infertility[9–11]. The requirement for faithful regulation of germline genes likely explains the engagement of several distinct chromatin marks in their repression, including DNAme[12], H3K9me3[13], and H2AK119ub1[14].

In contrast to the vast majority of CpG island (CGI) promoters, a subset of germline genes harbor CGI promoters that are de novo methylated between embryonic day (E) 4.5 and E6.5, yielding a high level of DNAme in the epiblast relative to the inner cell mass (ICM) of the blastocyst[3,4]. While the CGI promoter regions of these genes are demethylated in primordial germ cells (PGCs) coincident with the wave of genome-wide DNA demethylation, they remain hypermethylated in somatic tissues[15–17]. Indeed, 137 germline genes that acquire dense promoter DNAme in wild-type (WT) embryos by E6.5 are derepressed in *Dnmt3a/3b* double knock-out (DKO) embryos at E8.5[12]. This list of DNAme-sensitive (DMS) germline genes includes many that encode proteins involved in the repression of transposable elements (TEs), such as piRNA biogenesis factors. A number of these germline "genome-defence" (GGD) genes, including *Dazl*, *Ddx4*, *Mael*, *Mov10l1*, *Piwil2*, *Slc25a31*, *Taf7l*, and *Tex19.1*, were also previously reported to be regulated by promoter DNAme in E9.5 murine embryonic fibroblasts (MEFs)[18]. Similarly, Hill et al. identified an overlapping set of "germline reprogramming-responsive" genes that are upregulated in developing PGCs coincident with their demethylation[19]. Notably, a subset of these genes harbor E2F consensus (TCCCGC) motifs in their promoter regions and are bound by E2F6 in somatic cells[20,21]. While it has been speculated that DNMT3B recruitment through E2F6 transcriptional repressor mediates germline gene silencing in mouse somatic tissues[22], DNMT3A and DNMT3B act in a redundant manner to methylate the CGI promoters of such genes in peri-implantation embryos[4].

While DNAme in the promoter region of a subset of germline genes, including GGD genes, clearly plays a role in their repression post-implantation, many of the same genes are marked by H3K9me3 in ESCs. As these genes show loss of H3K9me3 and are upregulated in primed embryonic stem cells (pESCs) deficient in the lysine methyltransferase (KMTase) SETDB1, this histone post-translational modification (PTM) is also clearly involved in their silencing[13]. Indeed, H3K9me3 is bound by HP1 proteins and triple KO (TKO) of *Cbx1*, *Cbx3*, and *Cbx5* genes, which encode the three mammalian HP1 isoforms HP1β, HP1γ and HP1α, also leads to derepression of germline genes in pESCs[23]. While SETDB1-mediated silencing of TEs is dependent upon its canonical binding partner TRIM28/KAP1 and DNA-binding proteins of the KRAB-ZFP superfamily[24–28], evidence for a non-canonical SETDB1 recruitment pathway dependent upon the DNA-binding factor MAX was recently reported in pESCs[29].

MAX binds to E-box sites (CACGTG) when heterodimerized with its partner MGA[30], via an interaction between their basic helix-loop-helix leucine-zipper (bHLHZ) domains. Binding of this heterodimer represses the transcription of genes harboring the E-box motif, including many germline genes[31,32]. Consistent with the importance of the bHLHZ domain, MGA/MAX-mediated silencing of germline genes is alleviated in male germ cells at least in part by the expression prior to the onset of meiosis of a truncated form of MGA that lacks this domain[33]. Notably, the promoters of many germline genes harbor both E-box and E2F consensus motifs[32,34], indicating that the H3K9me3 and DNAme silencing pathways may converge at these genes. However, binding of MGA/MAX is inhibited by DNAme[35–37], suggesting that MGA/MAX-mediated silencing may occur prior to the wave of global de novo DNAme that occurs at implantation.

In addition to their reported roles in promoting H3K9me3 and DNAme, E2F6, MAX, and MGA are core subunits of the non-canonical PRC1 repressor complex PRC1.6[31,38–40], which is essential for embryonic viability[20,31,34,41–46]. PRC1.6 is abundant in pESCs[47] and recruited to germline genes, dependent upon MGA and/or E2F6[14,20,31,32,45,48–52]. Deletion of *Mga*, *Max*, or *E2f6* leads to derepression of an overlapping set of germline genes in pESCs or somatic cells[20,49,51], with loss of MGA in particular leading to dramatic upregulation[31,32]. Deletion of PRC1.6 subunits *L3mbtl2* and *Rybp*[45,50], as well as the associated H3K9 KMTase *G9a/Ehmt2*[45] also leads to upregulation of a partially overlapping set of germline genes in pESCs[34,53]. Furthermore, deletion of the genes encoding PRC1.6 catalytic subunits RING1A/B, which ubiquitylate H2A at lysine 119 (H2AK119ub1)[14], or PCGF6, which promotes RING1A/B catalytic activity[31,52], leads to upregulation of many PRC1.6-bound germline genes, including GGD genes. However, as pESCs are a population of mixed cellular differentiation states[54–56], it is difficult to determine the temporal order in which these silencing marks act or the role of interactions between them in mediating gene silencing. Of note, several of these germline genes are robustly derepressed in adult liver or brain in *Pcgf6* KO mice[34], indicating that PRC1.6 is required for silencing of such genes even in tissues in which their promoter regions are hypermethylated.

These observations indicate that repression of germline genes during mouse embryonic development is mediated at least in part by three epigenetic marks: DNAme, deposited by DNMT3A/3B; H3K9me3, deposited by SETDB1; and H2AK119ub1, deposited by PRC1.6 subunits RING1A/B. However, a number of questions concerning the specific developmental stages in which these silencing complexes are engaged and the interplay between them remain to be addressed. Furthermore, although the PRC2 complex, including JARID2, EED and the H3K27 KMTase EZH1/2, is recruited to PRC1-targeted genes, the role H3K27me3 plays in the silencing of germline genes has not been studied in detail[57,58].

Here, we characterize the dynamics of DNAme, H3K9me3, H3K27me3, and H2AK119ub1 in the early embryo, as well as in vitro using naïve ESCs (nESCs; cultured in 2i inhibitors PD0325901 and CHIR99021) to epiblast-like cell (EpiLC) to PGC-like cell (PGCLC) culture system. Consistent with the observations that MGA/MAX both recruit PRC1.6 and associate with H3K9me3[59,60], we find that H2AK119ub1 and H3K9me3 mark the promoter regions of GGD genes during pre-implantation stages, prior to their de novo DNAme. Furthermore, KO of core/catalytic subunits of each of the previously implicated repressive complexes in nESCs and EpiLCs reveals that PRC1.6 subunits MGA and PCGF6 are required for SETDB1-mediated H3K9me3 at bound GGD genes. Finally, consistent with their dynamics in vivo, we demonstrate that H2AK119ub1 and H3K9me3 are generally more important for silencing of GGD genes than DNAme in nESCs, while DNAme plays a more pronounced role in EpiLCs. Taken together, these

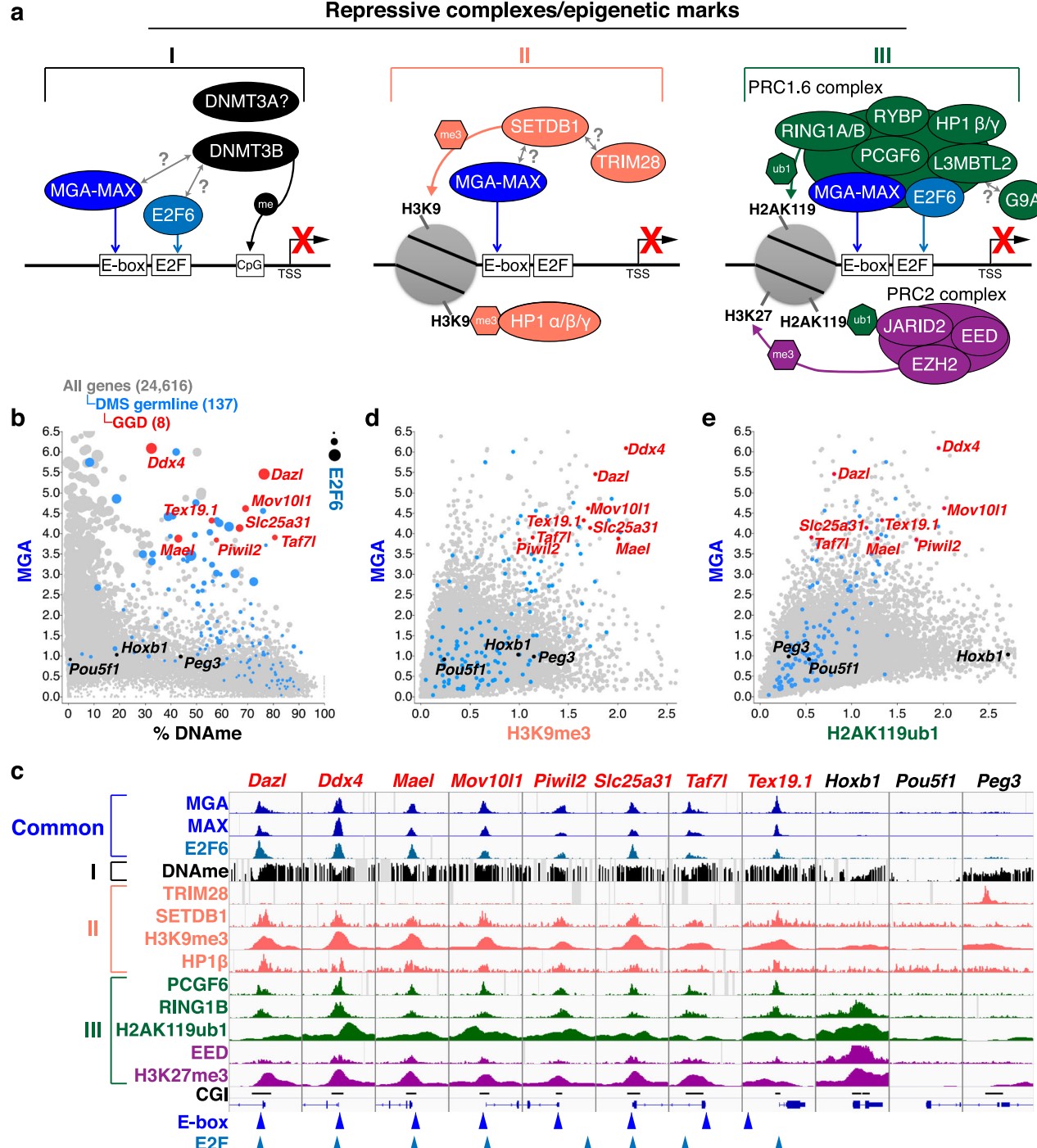

observations suggest that PRC1.6 and SETDB1 are essential for repression of GGD genes during pre-implantation embryonic development, prior to the establishment of DNAme, while DNAme is engaged in silencing of these genes only after implantation.

## Results

**Roles of DNAme, H3K9me3, and H2AK119ub1 in repression of germline genes in pESCs.** Three distinct epigenetic marks, including DNAme, H3K9me3, and H2AK119ub1, deposited by de novo DNMT3A/3B, SETDB1, and PRC1.6, respectively, have been previously implicated in silencing of germline genes in pESCs and/or somatic cells in mice (Fig. 1a). To determine whether the 137 DMS germline genes upregulated in E8.5

$Dnmt3a/3b$ DKO embryos[12] (Supplementary Data 1) are hyper-methylated as well as bound by SETDB1 and PRC1.6 in pESCs, we reanalyzed a series of published whole-genome bisulfite sequencing (WGBS) and ChIP-seq datasets (Supplementary Data 2). As expected, the CGI promoter regions of most DMS germline genes, including all GGD genes[18], are methylated in E8.5 embryos as well as pESCs (Fig. 1b, c and Supplementary Fig. 1a). However, not all of the 137 DMS germline genes are enriched for MGA/MAX and/or E2F6 (Fig. 1b, c and Supplementary Fig. 1b), consistent with the fact that 82 and 97 of these genes include E-box or E2F motifs, respectively, in their promoter regions, with only 61 including both motifs (Supplementary Fig. 1c and Supplementary Data 1). Manual curation of the function of the 40 genes in ESCs enriched for both MGA

**Fig. 1 Repressive complexes and associated epigenetic marks acting on DMS germline genes in pESCs. a** Three repressive complexes (I–III) which act on a subset of germline genes have been identified previously in primed (serum-grown) ESCs (pESCs) and/or somatic cells. The promoter regions of these genes are bound by heterodimers of the transcription factors MGA and MAX, which are recruited to E-Box sites as well as E2F6, which binds to E2F sites. Complex I: The de novo DNA methyltransferase DNMT3B is recruited to germline genes via MGA/MAX and/or E2F6. Complex II: The lysine (K)MTase SETDB1, which deposits H3K9me3, is recruited to germline genes dependent upon MAX. SETDB1 is known to interact with the co-repressor TRIM28. HP1 proteins are recruited to H3K9me3 marked regions. Complex III: The non-canonical PRC1 complex PRC1.6, including PCGF6, L3MBTL2, RYBP, HP1β/γ, and RING1A/B subunits, is recruited to germline genes by MGA/MAX and E2F6. RING1A/B deposit H2AK119ub1, which in turn is recognized by the PRC2 complex, promoting H3K27me3 deposition. The KMTase G9A/EHMT2 was previously reported to associate with PRC1.6 complex via L3MBTL2[45]. **b** Scatterplot showing the relationship between the enrichment (RPKM) of MGA or E2F6 (dot size) and % DNAme (TSS −0.9/+0.4 kb). **c** Screenshots of ChIP-seq and WGBS tracks showing RPM values of the indicated subunits of each of the complexes described in (**a**) as well as histone PTMs and % DNAme at the TSSs (±3 kb) of genes in pESCs. E-Box motifs and E2F motifs are shown as arrowheads. Annotated CpG islands (CGIs) are also shown. **d**, **e** Scatterplots showing the relationship between (**d**) MGA and H3K9me3, or (**e**) MGA and H2AK119ub1 (RPKM), around the TSSs (±2 kb) of all annotated genes in pESCs. In panels (**b**, **d**, **e**), all genes are labeled in (gray) while the 137 DMS germline genes upregulated in *Dnmt3a/3b* DKO embryonic day 8.5 embryos[12] and subset categorized as germline "genome-defence" (GGD) genes (also upregulated in DNAme-deficient MEFs[18]) are labeled in blue and red, respectively. Control genes *Hoxb1* (a canonical PRC1/2 target), *Pou5f1* (an actively transcribed gene), and *Peg3* (a TRIM28 target) are labeled in black. See Supplementary Data 2 for details of published datasets analyzed.

(RPKM > 2) and E2F6 (RPKM > 1) and showing >30% DNAme in the promoter region revealed 18 associated with meiosis and 16 associated with genome-defence/piRNA biogenesis (Supplementary Data 3), including all GGD genes. Intriguingly, the DNAme level of these MGA- and E2F6-bound genes is generally lower than that of the DMS germline genes that are not bound by these transcription factors (TFs), a trend also observed in E8.5 embryos (Fig. 1b and Supplementary Fig. 1a).

Many of the DMS germline and all GGD genes are also enriched for H3K9me3, SETDB1 and HP1β (Fig. 1c, d and Supplementary Fig. 1c, d), as well as H2AK119ub1 and PRC1.6 complex subunits PCGF6 and RING1B (Fig. 1c, e and Supplementary Fig. 1c, e). These regions are also enriched for H3K27me3 and the PRC2 complex subunit EED (Supplementary Fig. 1f), likely reflecting H2AK119ub1-dependent recruitment of PRC2[14,61]. TRIM28, on the other hand, showed only low levels of enrichment at DMS germline gene promoters, including those with high H3K9me3 (Fig. 1c and Supplementary Fig. 1d), consistent with a non-canonical mechanism of SETDB1 recruitment to these loci. Indeed, co-enrichment of H2AK119ub1 and H3K9me3 is relatively rare at genic promoters (Supplementary Fig. 1g). Thus, PRC1.6-bound genes, including GGD genes, are marked by an unusual combination of repressive histone PTMs and intermediate levels of DNAme in pESCs.

To determine whether these repressive complexes play a role in silencing of DMS germline (and in particular GGD) genes in pESCs, we next reanalyzed a series of published RNA-seq data (Supplementary Data 2), where key subunits of each of the complexes described above were depleted. Strikingly, DMS GGD genes as well as other methylated germline genes bound by MGA and/or enriched for H3K9me3 or H2AK119ub1 were derepressed upon KO or knock-down (KD) not only of *Mga*, *Max*, and *Dnmt3a/3b*, but also of *Setdb1*, *Cbx1/3/5*, and *Pcgf6* (Fig. 2a–d and Supplementary Fig. 2a–d). These observations are consistent with previous studies showing that a subset of the DMS germline genes, including GGD genes, are derepressed in pESCs following KO of *L3mbtl2*, another core PRC1.6 subunit[40,45,62]. To determine whether H2AK119ub1 is required for silencing, we generated a *Ring1b* conditional (c)KO line. GGD genes in particular were upregulated following *Ring1b* deletion in pESCs (Fig. 2e), to levels similar to those observed in *Pcgf6* cKO pESCs (Fig. 2d and Supplementary Fig. 2c). In contrast, only modest upregulation of GGD genes was observed in *Trim28* cKO pESCs, consistent with the absence of TRIM28 binding in the promoter regions of these genes (Supplementary Fig. 2b). Intriguingly, while several DMS germline genes were upregulated in *G9a* KO pESCs, GGD genes were only modestly upregulated or showed no

change in expression in these cells (Supplementary Fig. 2c). Furthermore, no upregulation of DMS germline genes was observed in *Eed* cKO ESCs (Supplementary Fig. 2d), indicating that H3K27me3, while enriched at PRC1.6-bound promoters, is not required for silencing of these genes. Taken together, these analyses indicate that in pESCs, repression of a subset of DMS germline genes is dependent not only upon DNAme, but also on MGA/MAX, SETDB1, and the canonical PRC1.6 complex. However, as pESCs are a mixed population of cells at distinct differentiation states[54–56], it is not possible to deduce either the temporal order in which these silencing complexes act during development, or their interdependencies in mediating such silencing.

**H2AK119ub1, H3K9me3, and H3K27me3 are sequentially deposited prior to de novo DNAme at DMS germline genes.** To investigate the dynamics of these histone PTMs as well as DNAme at DMS germline genes during early embryonic/PGC development, we analyzed previously published ChIP-seq and WGBS datasets from 2-cell, morula, ICM, and epiblast as well as PGCs (Fig. 3a, b). In the promoter regions of many DMS germline gene loci, H2AK119ub1, H3K9me3, and H3K27me3 are already present in the ICM or earlier stages (Fig. 3b and Supplementary Fig. 3a, b) and H2AK119ub1 is clearly deposited at DMS germline and GGD genes as early as the 2-cell stage, followed by H3K9me3 in morula and ICM and H3K27me3 in ICM and E6.5 epiblast[63,64] (Supplementary Fig. 3a, b). In contrast, DNAme is not deposited at these loci until the epiblast stage, as expected (Fig. 3b). As observed in pESCs, MGA- and E2F6-bound genes show a lower level of DNAme in E6.5 epiblast than most other DMS germline genes (Supplementary Fig. 3c). Despite the increase in repressive marks in epiblast, expression of several GGD genes transiently increases in epiblast relative to ICM, perhaps due to the simultaneous upregulation of positive regulatory factors that bind to their CGI promoters (Fig. 3c). To characterize the dynamics of H3K9me3 after implantation, we carried out H3K9me3 ULI-ChIP-seq[65] for E9.5 PGCs isolated by FACS (Supplementary Fig. 3d). While H3K9me3, like H3K27me3, persists at DMS germline genes in the epiblast, both H3K9me3 and H3K27me3 progressively decrease at these genes in PGCs, coincident with their DNA demethylation and upregulation in E13.5 PGCs (Fig. 3b, c and Supplementary Fig. 3b). Thus, H2AK119ub1 and H3K9me3 are deposited prior to de novo DNAme at many DMS germline gene promoters, including all GGD genes, implicating DNAme-independent repression of these loci during pre-implantation development.

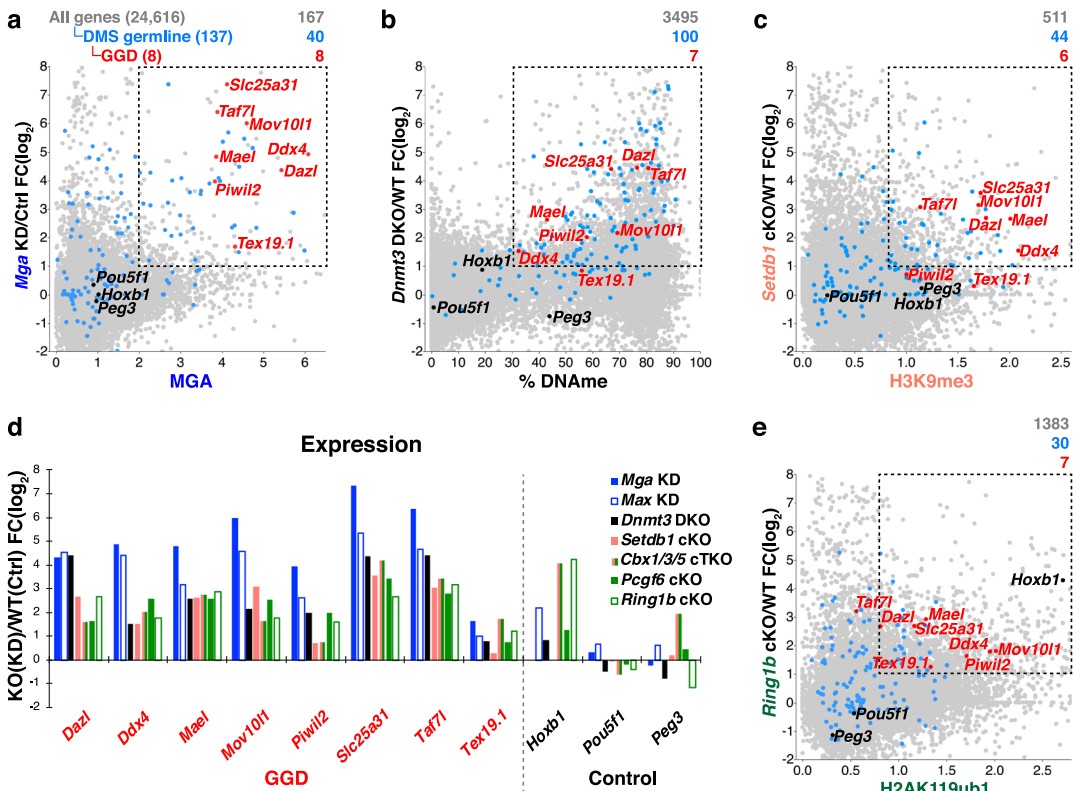

**Fig. 2 Depletion of key repressive complex subunits impacts expression of DMS germline genes in pESCs. a–c** Scatterplots showing the fold-change (FC) of gene expression in knock-out (KO) or knock-down (KD) vs WT/control (Ctrl) pESCs (y-axis) for: **a** *Mga* KD versus MGA enrichment (RPKM, TSS ±2 kb), **b** *Dnmt3a/3b* Double KO (*Dnmt3* DKO) vs % DNAme (TSS −0.9/+0.4 kb), **c** *Setdb1* conditional KO (cKO) vs H3K9me3 enrichment (TSS ±2 kb). For (**a–c**), ChIP-seq data are presented as RPKM values. **d** Bar graph showing the FC of gene expression in repressor complex subunit KO/KD pESCs relative to WT/Ctrl at GGD and control genes. **e** Scatterplot showing the FC of gene expression in *Ring1b* cKO vs H2AK119ub1 enrichment (RPKM, TSS ±2 kb). The total number of genes in each category (in parentheses), as well as the number of genes showing a >2-fold increase in expression and MGA enrichment (RPKM > 2), DNAme level >30%, H3K9me3 enrichment (RPKM > 0.8), and H2AK119ub1 enrichment (RPKM > 0.8), respectively, is shown at the top of each plot. All, DMS germline and GGD genes are color-coded as in panel (**a**).

To further characterize the roles of these chromatin marks, we employed the PGCLC culture system[66], in which nESCs, EpiLCs, day (d) 4–6 PGCLCs and d4 plus 7 days of culture (d4c7) PGCLCs represent the in vitro counterparts of E3.5 ICM, E5.5-6.5 epiblast, E9.5-11.5 PGCs, and E13.5 PGCs, respectively (Fig. 3a and Supplementary Fig. 3a, d). As expected, while DNAme levels in nESCs are relatively low in the promoter regions of DMS germline genes, H2AK119ub1, H3K9me3, and H3K27me3 are clearly enriched at a subset of these genes, including GGD genes (Fig. 3b and Supplementary Fig. 3a, b). Consistent with PRC1.6 binding at their promoter regions, the enrichment of H2AK119ub1 in nESCs and EpiLCs is positively correlated with the enrichment levels of MGA and E2F6, as measured in pESCs and nESCs, respectively (Supplementary Fig. 3e).

Following EpiLC induction, these loci are de novo methylated and retain each of these histone PTMs, with enrichment modestly increased relative to nESCs. Mirroring the E6.5 epiblast, DNAme levels in EpiLCs are still lower at MGA- and E2F6-bound genes (including GGD genes) relative to most other genes, including DMS germline genes that are not bound by these TFs (Supplementary Fig. 3c). These observations indicate that PRC1.6-bound promoters are refractory to the wave of de novo DNAme that occurs during epiblast formation. H2AK119ub1, H3K9me3, and H3K27me3 are subsequently reduced in d4-6 PGCLCs, while DNAme persists (Fig. 3b and Supplementary Fig. 3a, b). As observed in the epiblast, despite the increase in repressive marks in EpiLCs, expression of the majority of the GGD genes transiently increases in EpiLCs relative to nESCs

(Fig. 3c). Regardless, these genes are then downregulated in d4 PGCLCs, before their upregulation again in d4c7 PGCLCs, as expected. Based on these observations, we subsequently employed the nESC/EpiLC/PGCLC system to model the relative importance of and crosstalk between each repressive pathway during pre- and post-implantation development.

**DNAme plays a more important role in silencing of DMS germline genes in EpiLCs than nESCs.** Analysis of previously generated WGBS data from DNMT-deficient E6.5 epiblast cells[67] revealed a dramatic reduction in DNAme at DMS germline genes in both *Dnmt1* KO and *Dnmt3a/3b* DKO embryos (Fig. 4a, b), confirming the importance of de novo DNAme at this stage. As observed in the transition from ICM to E6.5 epiblast to E8.5 embryo, EpiLCs and the extended culture of EpiLCs (exEpiLCs) also show progressively higher levels of DNAme in the promoter regions of DMS germline genes, relative to the nESCs from which they are derived[68–70]. However, nESCs show a modestly higher level of DNAme in these regions than E3.5 ICM (Fig. 4a, b). While nESCs express lower levels of *Dnmt3a*, *Dnmt3b*, and *Uhrf1* than pESCs[68,69], they do express TET1, which mediates oxidation of 5-methylcytosine (5mC) to 5-hydroxymethylcytosine (5hmC) in an active DNA demethylation pathway[70]. As bisulfite sequencing cannot discriminate between 5mC and 5hmC, we surmised that the DNAme observed in DMS germline genes in nESCs may reflect at least in part the presence of 5hmC. Analysis of previously published 5hmCIP-seq data[71] revealed that relative to all other genes in nESCs,

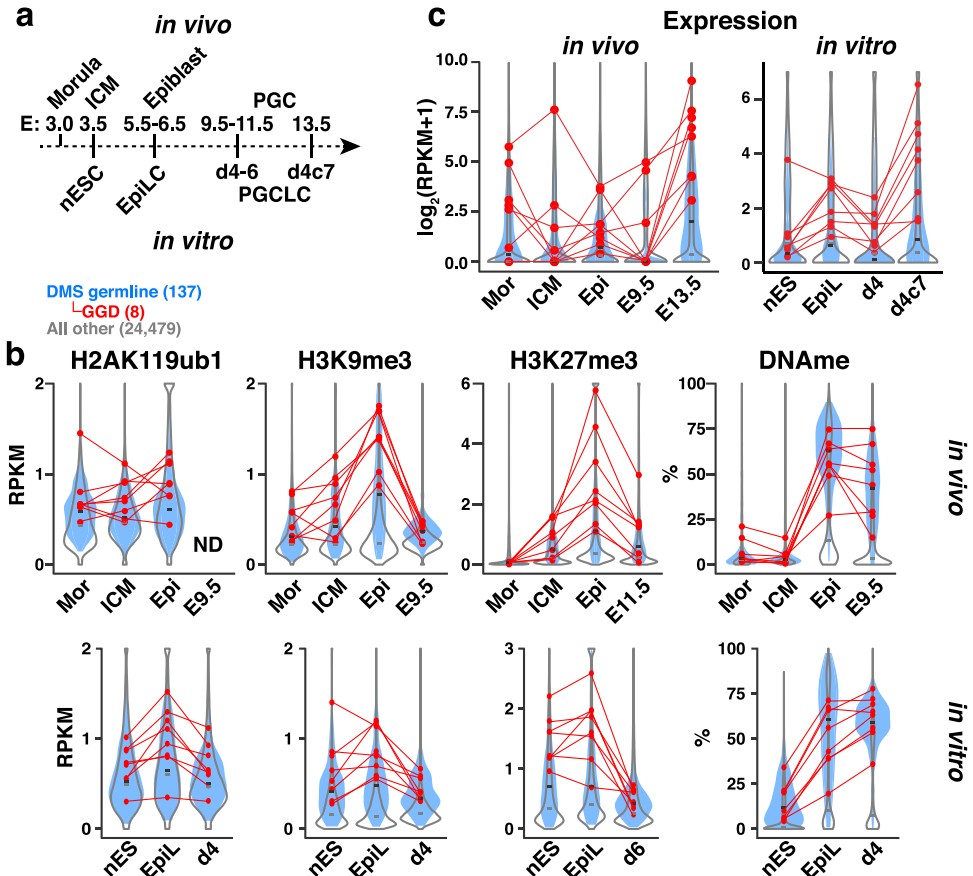

**Fig. 3 H2AK119ub1 and H3K9me3 precede DNAme at DMS germline genes during embryonic development and EpiLC induction. a** Application of an in vitro differentiation system to model pre-implantation (ICM), post-implantation (epiblast) and primordial germ cell (PGC) development. Naïve (n) ESCs (cultured in 2i), EpiLCs, day (d) 4–6 and d4 plus 7 days of culture (d4c7) PGCLCs correspond to embryonic day (E) 3.5 inner cell mass (ICM), E5.5-6.5 epiblast (Epi), E9.5-11.5, and E13.5 PGCs, respectively. **b** Violin plots showing in vivo and in vitro dynamics of H2AK119ub1, H3K9me3, and H3K27me3 enrichment (TSS ±2 kb) as well as DNAme levels (TSS −0.9/+0.4 kb) at 137 DMS germline (blue filled), 8 GGD (red data points) and all other gene loci (open). ND = no data. CUT&RUN and ChIP-seq data are presented as RPKM values. Epi = E6.5 epiblast. **c** Violin plots showing gene expression ($\log_2(\text{RPKM} + 1)$) during embryonic development (in vivo) and in vitro, labeled as shown in panel (**a**). Epi = E5.5 epiblast. See Supplementary Data 2 for details of published datasets analyzed.

the promoters of DMS germline genes, including GGD genes, are indeed enriched for 5hmC (Supplementary Fig. 4a, b). Surprisingly, the levels of 5hmC enrichment are similar between nESCs and EpiLCs, and analysis of previously published ChIPseq data also reveals similar levels of TET1 enrichment between these cell types[71,72]. Taken together with the fact that WGBS indicates significantly higher levels of DNAme in EpiLCs, these data suggest that the ratio of 5hmC-to-5mC is higher in nESCs. Closer inspection of these loci reveals that 5hmC and TET1 peaks overlap with each other and with MGA/MAX and E2F6 in the promoter region, corresponding to regions of reduced DNAme relative to flanking CpGs (Supplementary Fig. 4c). Thus, the low levels of DNAme observed in the promoter regions of GGD genes in nESCs actually represents the presence of both 5mC and 5hmC, likely reflecting reiterative de novo DNAme followed by oxidation of this mark under these culture conditions.

To directly address the role of DNAme in silencing of DMS germline genes in nESCs, we employed a line harboring floxed alleles of *Dnmt1, Dnmt3a*, and *Dnmt3b*[73]. Following the addition of 4-hydroxytamoxifen (4OHT) for 4 days, RNA-seq was conducted on control or *Dnmt1/3a/3b* conditional triple (cT)KO nESCs, EpiLCs (isolated after 2 days of differentiation) (Fig. 4c). As expected, given the low levels of DNAme in nESCs, only 14/137 DMS germline genes showed >30% promoter DNAme in control

nESCs and were upregulated in *Dnmt* cTKO nESCs (Fig. 4d). The observed upregulation of the GGD genes *Dazl* and *Piwil2*, as well as the control gene *Hoxb1* (Fig. 4d, e), is likely an indirect effect, as each of these genes shows very low levels of DNAme in nESCs. In contrast, 74/137 DMS germline genes showed >30% promoter DNAme in control EpiLCs and were upregulated in *Dnmt* cTKO EpiLCs (Fig. 4d). These include GGD genes *Slc25a31, Taf7l*, and *Tex19.1*, which were only modestly upregulated in DNAme-deficient nESCs (Fig. 4e). Intriguingly, the remaining GGD genes *Ddx4, Mael*, and *Mov10l1*, which ranged in promoter DNAme levels from 5.5% to 21.3% and 19.5% to 67.3% in WT nESCs and EpiLCs, respectively, remained repressed following *Dnmt* cTKO in both cultures. Thus, consistent with the wave of global de novo DNAme that takes place with EpiLC induction, DNAme plays a more important role in silencing of DMS germline genes in EpiLCs than nESCs, including several GGD genes. However, a subset of GGD genes is silenced independent of DNAme even in EpiLCs.

## PRC1.6 plays a more important role than DNAme in silencing of GGD genes in nESCs.
To investigate the potential crosstalk between H3K9me3, H2AK119ub1, and DNAme in the regulation of DMS germline genes, we carried out ULI-ChIP-seq on *Setdb1* cKO, *Pcgf6* cKO, and *Dnmt* cTKO nESCs and EpiLCs. In parallel,

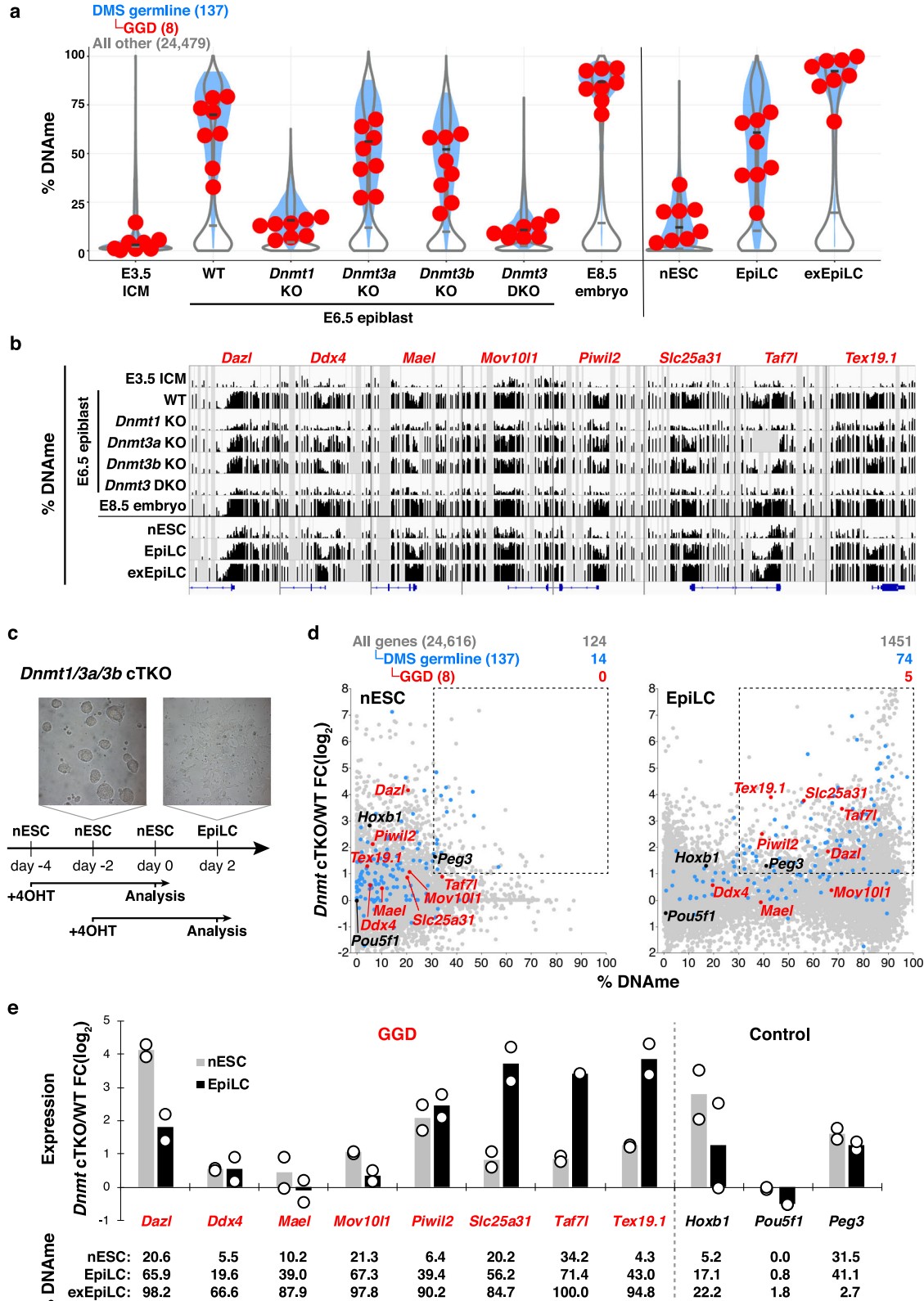

we analyzed *Mga-ΔHLH* nESCs and EpiLCs, which express a truncated form of the protein from the native locus that lacks the bHLHZ domain and in turn cannot bind to E-box sites[74]. As conditional KO of *Setdb1* in nESCs leads to a dramatic reduction in viability by day 3 in culture[75], cKO lines of *Setdb1* and *Pcgf6* were harvested following 2 days of 4OHT treatment. RT-qPCR and western blotting confirmed that both mRNAs and proteins

are lost by this time point (Supplementary Fig. 5a–c). H3K9me3 levels were reduced at DMS germline and GGD genes in both nESCs and EpiLCs in each of the mutants analyzed (Fig. 5a), with the effect more pronounced for the *Mga-ΔHLH*, *Setdb1* cKO, and *Pcgf6* cKO lines than the *Dnmt* cTKO line. Thus, both the bHLHZ DNA-binding domain of MGA and the PCGF6 core subunit of PRC1.6 are required for SETDB1 recruitment and/or

**Fig. 4 DNAme plays a more important role in silencing of DMS germline genes in EpiLCs than nESCs. a** Violin plots showing the distribution of DNAme levels in the promoter regions (TSS −0.9/+0.4 kb) of the 137 DMS germline (blue filled), 8 GGD (red data points) and all other gene loci (open) in E3.5 ICM, WT, *Dnmt1-, Dnmt3a-, Dnmt3b-,* or *Dnmt3a/3b* double KO (DKO) E6.5 epiblast cells and E8.5 embryos. Data for nESCs, EpiLCs, and exEpiLCs are also shown. **b** Genome browser tracks showing % DNAme in the promoter regions (TSS ±3 kb) of GGD gene loci at the developmental time points shown in panel (**a**). For each WGBS track, regions highlighted in gray reflect the absence of DNAme data. **c** A schematic representation of the induction of *Dnmt1/3a/3b* conditional triple KO (cTKO) in nESCs and derived EpiLCs. To induce *Dnmt* cTKO, 4-hydroxytamoxifen (4OHT) was added at the indicated time points and cells were harvested for RNA-seq analysis at the time points indicated. **d** Scatterplots showing the fold-change (FC) of gene expression in *Dnmt* cTKO nESCs (*n* = 2) and EpiLC (*n* = 2) vs % DNAme in the promoter region (TSS −0.9/+0.4 kb). All, DMS germline and GGD genes are color-coded as shown and the number of genes in each category showing % DNAme >30 and a concomitant >2-fold increase in expression is included at the top right of each plot. **e** Bar graph showing the mean FC of expression of GGD genes (red) and control genes (black) in nESCs and EpiLCs with *Dnmt* cTKO relative to WT cells. Data points show biological duplicates. % DNAme in nESCs, EpiLCs and exEpiLCs in the promoter regions (TSS −0.9/+0.4 kb) of each gene are also shown.

activity at these loci. While H2AK119ub1 at DMS germline and GGD genes was also significantly reduced in *Pcgf6* cKO nESCs and EpiLCs, as expected, this mark was only modestly reduced in the *Dnmt* cTKO and unaffected in the *Mga-*ΔHLH and *Setdb1* cKO lines (Fig. 5b). This suggests that neither MGA bHLHZ-dependent E-box-binding nor SETDB1 is required for persistence of H2AK119ub1 at these loci, at least at the time points analyzed.

RNA-seq analyses of these KO nESCs and EpiLCs revealed varying levels of derepression of DMS germline genes (Fig. 5c and Supplementary Data 4). *Mga-*ΔHLH showed the most consistent upregulation of GGD genes in nESCs and EpiLCs (Fig. 5d). *Dnmt* cTKO in nESCs resulted in only modest upregulation of most GGD genes when compared to *Mga-*ΔHLH and *Pcgf6* cKO nESCs. For several of these genes, DNAme is apparently dispensable for silencing in EpiLCs as well, whereas PRC1.6 is required for transcriptional repression in both nESCs and EpiLCs. However, for 3 of the GGD genes, namely *Slc25a31, Taf7l,* and *Tex19.1,* a more robust upregulation is clearly evident in *Dnmt* cTKO EpiLCs than nESCs. Consistent with this trend, DESeq2-based TCC (Tag Count Comparison) analysis[76] of *Dnmt* cTKO EpiLCs yielded 70 upregulated DMS germline genes, versus only 28 in *Dnmt* cTKO nESCs (Supplementary Data 4). The increased level of DNAme in the CGI promoter regions of these genes (Fig. 4e) is likely responsible for the greater impact of *Dnmt* cTKO in EpiLCs. Furthermore, the engagement of DNAme-mediated repression may explain the reduced impact of *Pcgf6* deletion on expression of all of the GGD genes in EpiLCs relative to nESCs (Fig. 5d).

As *Pcgf6* deletion disrupted both H2AK119ub1 and H3K9me3 deposition, we next wished to determine the relative importance of these histone PTMs in transcriptional repression. We carried out RNA-seq on *Setdb1* cKO, *Ring1b* cKO, as well as *Setdb1/Ring1b (Set/Ring)* cDKO nESCs and EpiLCs, as above (Supplementary Fig. 5a). DESeq2-based TCC analysis yielded a similar number of upregulated DMS germline genes in *Ring1b* cKO vs *Pcgf6* cKO nESCs or EpiLCs (Supplementary Fig. 5c and Supplementary Data 4). However, DMS germline and GGD genes showed a greater level of derepression in *Setdb1/Ring1b* cDKO than individual *Ring1b* or *Setdb1* cKO nESCs and EpiLCs (Fig. 5c, d and Supplementary Fig. 5d, e), with the fold-increase in expression for many GGD genes in *Setdb1/Ring1b* cDKO nESCs similar to that observed in *Pcgf6* cKO nESCs (Fig. 5d and Supplementary Fig. 5e). These results suggest that RING1B and SETDB1 act in a combinatorial manner to repress PRC1.6-bound genes and are consistent with the observation that loss of PCGF6 disrupts both H2AK119ub1 and H3K9me3 deposition. However, unlike the *Pcgf6* cKO, *Setdb1/Ring1b* cDKO EpiLCs show a level of derepression that does not decrease relative to nESCs, indicating that SETDB1 is still essential for robust silencing of these genes even in the presence of a higher level of DNAme in their promoter regions.

Intriguingly, the fold-change in expression of some GGD genes in *Mga-*ΔHLH EpiLCs and exEpiLCs far exceeds that observed in EpiLCs derived from the other KO lines, including *Setdb1* cKO (Fig. 5d and Supplementary Fig. 6a), indicating that MGA plays a particularly important role in repression of GGD gene expression. Notably, the transcription factor MEIOSIN was recently reported to play a critical role in driving expression of germline genes, including GGD genes, in testis[77]. As this gene shows greater upregulation in *Mga-*ΔHLH nESCs/EpiLCs than any of the other KO lines analyzed (Supplementary Fig. 6a), the relatively high level of derepression of GGD genes in *Mga-*ΔHLH cells may be due to the loss of PRC1.6 and SETDB1 binding combined with increased binding of MEIOSIN at their promoter regions.

To determine whether MGA binding is required for de novo DNAme, we isolated genomic DNA from *Mga-*ΔHLH EpiLCs and exEpiLCs and conducted bisulfite sequencing on the CGI promoter regions of *Mael* and *Mov10l1*. DNAme levels were significantly lower at both genes relative to control EpiLCs, as observed in *Dnmt* cTKO EpiLCs (Fig. 5e and Supplementary Fig. 7a, b). Since these genes are not derepressed in *Dnmt* cTKO EpiLCs (Figs. 4e and 5d), these observations indicate that even when present, DNAme is not necessarily required for silencing of GGD genes in EpiLCs. Only a moderate decrease or no change in DNAme was observed in these regions in *Setdb1-, Pcgf6-,* or *Ring1b* cKO EpiLCs (Supplementary Fig. 6a, b), indicating that H3K9me3 and H2AK119ub1 may play redundant roles in potentiating de novo DNAme of these loci. Importantly, while control exEpiLCs showed a further gain of DNAme, as expected, DNAme levels at both of these loci remained significantly lower in *Mga-*ΔHLH exEpiLCs (Supplementary Fig. 7a, b). The apparent failure of de novo DNAme is consistent with the dramatic upregulation of both genes in *Mga-*ΔHLH exEpiLCs (Supplementary Fig. 6b).

To directly address whether PRC1.6 is indeed required for de novo DNAme of DMS germline gene promoter regions in vivo, we reanalyzed published WGBS data from *Ring1b, L3mbtl2,* and *G9a* KO E6.5 epiblasts[67]. Though not to the extent observed in *Dnmt3a/3b* DKO E6.5 epiblast, specific DMS germline genes showed decreased DNAme in each of the other KOs relative to controls (Fig. 5f). This includes the promoter regions of a subset of GGD genes, with the exception of *Dazl*, which retains high levels of DNAme in all mutants (Fig. 5g). As the decrease in DNAme was least pronounced in the *G9a* KO, we also analyzed previously published RRBS data from *G9a* KO E8.5 embryos. Strikingly, relative to control embryos at this stage, little change in DNAme was observed in the promoter regions of DMS germline genes, including GGD genes (Supplementary Fig. 8a–c), indicating that G9A is dispensable for further accumulation of DNAme at those regions after epiblast formation. Finally, analysis of data published while this paper was under review reveals that several

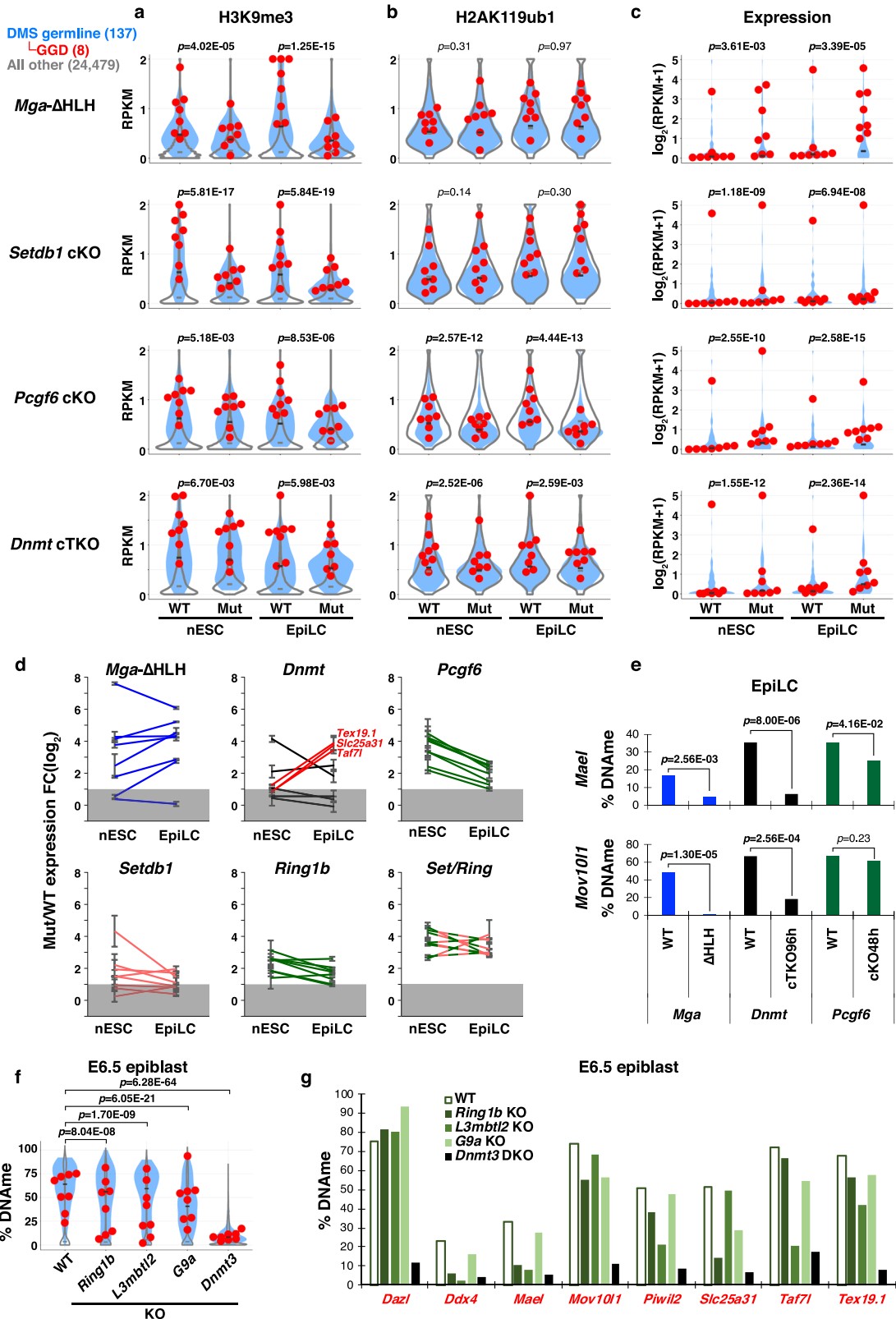

GGD genes, including *Mael*, are upregulated in embryos deficient in E2F6[78]. While DNAme in the promoter regions of several DMS germline genes with E2F motifs is significantly decreased in *E2f6* KO E8.5 embryos[78], only a modest reduction or no change was apparent at GGD genes, including *Mael* (Supplementary Fig. 8b, c). As all GGD genes harbor both E2F and E-box motifs, these observations indicate that E2F6 may be less important for de novo DNAme than MGA/MAX at genes harboring both motifs.

## Discussion

By integrating numerous previously published datasets from pre-implantation stages, we show that the promoter regions of DMS germline genes bound by MGA/MAX and E2F6 in ESCs show

**Fig. 5 Roles of MGA, SETDB1, and PRC1.6 in silencing and DNAme of DMS germline genes in nESCs and EpiLCs. a–c** Violin plots showing the enrichment (RPKM) of H3K9me3 (TSS ±1 kb) and H2AK119ub1 (TSS ±2 kb) at 137 DMS germline (filled), 8 GGD (red data points) and all other gene loci (open) (**a**, **b**), as well as their expression levels ($\log_2(\text{RPKM} + 1)$) ($n = 2$) (**c**) in control (WT) nESCs and EpiLCs vs mutants (Mut) of the indicated repressor complex subunits. Two-tailed paired-samples $t$-tests were performed for each mutant/cKO and WT pair of all DMS germline gene values. To induce *Setdb1* cKO or *Pcgf6* cKO, 4OHT was added for 48 h during ESC(2i) culture or EpiLC induction (cKO48h). **d** Line graphs showing the FC ($\log_2$) in expression of GGD genes in nESCs and EpiLCs for each of the mutant/cKO lines indicated. Gray area represents FC < 2. Error bars show SE of biological replicates. **e** Bar graphs showing the levels of DNAme in the promoter regions of *Mael* and *Mov10l1* in *Mga*-ΔHLH, *Dnmt* cTKO, or *Pcgf6* cKO vs WT EpiLCs (see Supplementary Fig. 6). Two-tailed Mann-Whitney $U$-tests were performed between mutant and WT. **f** Violin plots showing DNAme profiles of the promoter regions (TSS ±0.3 kb) of DMS germline, GGD and all other genes (labeled as in panel **a**) in WT, *Ring1b*-, *L3mbtl2*-, or *G9a* KO E6.5 epiblast cells. Two-tailed paired-sample $t$-tests were performed for each KO and WT pair of all DMS germline gene values. **g** Graph showing the levels of DNAme in the promoter regions (TSS ±0.3 kb) of GGD genes in WT vs *Ring1b*-, *L3mbtl2*-, or *G9a* KO E6.5 epiblast cells.

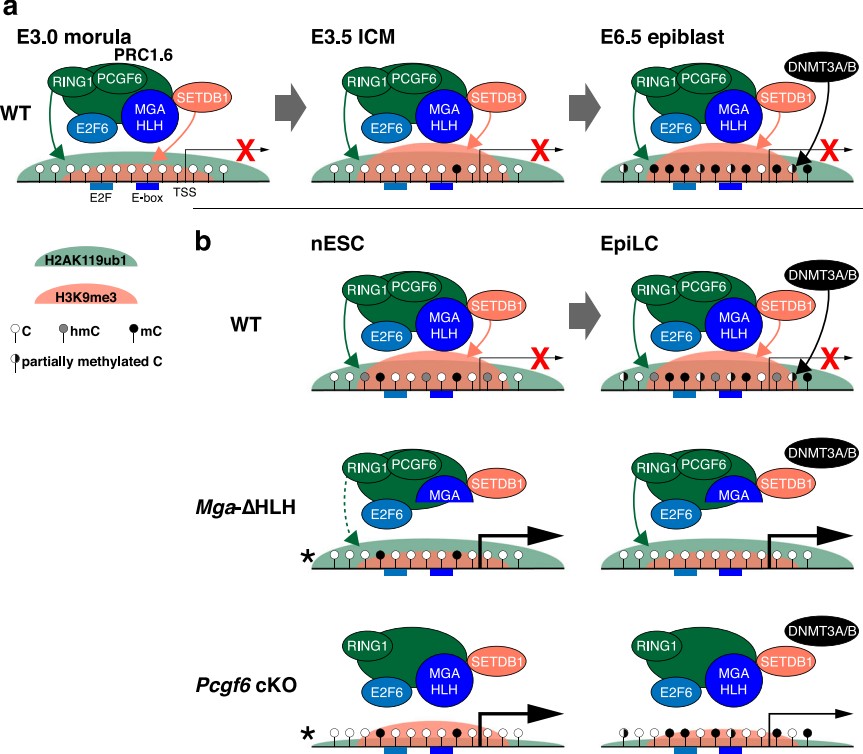

**Fig. 6 The order and crosstalk among repressive mechanisms acting at GGD genes during embryogenesis. a** The CGI promoter regions of GGD genes with E2F and E-box motifs are bound by MGA and E2F6 (as determined in ESCs). Prior to implantation (E3.5 ICM), these GGD genes are sequentially marked by H2AK119ub1, dependent in part upon PRC1.6, H3K9me3, dependent upon SETDB1 and H3K27me3, dependent upon PRC2 (not shown). DNMT3A/3B-dependent de novo DNAme occurs at these regions only after implantation (~E6.5 epiblast) and accumulates further thereafter. Single-nucleotide resolution data for hmC has not been determined in ICM or epiblast. MGA/PRC1.6 binding may be disrupted once the CpG site embedded within the E-box motif is methylated (not shown), leaving DNAme as the only repressive mark. **b** In vitro modeling of this transition recapitulates H3K9me3 and H2AK119ub1 enrichment, as well as de novo DNAme of these promoter regions following induction of EpiLCs from nESCs. While nESCs show lower levels of DNAme than EpiLCs, hmC levels at GGD gene promoters are detected at similar levels in both cell types, likely reflecting TET1 activity (not shown). The absence of the bHLHZ DNA-binding domain of MGA (*Mga*-ΔHLH) is associated with a reduction of H3K9me3 at these loci in nESCs and disruption of de novo DNAme following EpiLCs induction. PCGF6, a subunit of PRC1.6 is required for efficient deposition of H3K9me3 as well as H2AK119ub1 in nESCs and EpiLCs, but its deletion has only a modest impact on de novo DNAme in EpiLCs relative to *Mga*-ΔHLH. Germline genes are derepressed to a greater extent in *Pcgf6* cKO nESCs than EpiLCs, perhaps due to the accumulation of DNAme in the latter. *The status of mC and hmC of GGD gene promoters was not determined in *Mga*-ΔHLH or *Pcgf6* cKO nESCs. *Setdb1* and *Ring1b* Double KO also leads to robust derepression of GGD genes in nESCs (not shown), confirming the importance of both H3K9me3 and H2AK119ub1 in their transcriptional repression prior to the onset of de novo DNAme.

sequential accumulation of H2AK119ub1, H3K9me3, and H3K27me3, followed by de novo DNAme in the epiblast (Fig. 6a). The deposition of these histone PTMs prior to the establishment of DNAme is consistent with roles for PRC1.6 and SETDB1 prior to and independent of DNAme-mediated silencing. To study the relative importance of these repressive marks, we utilized the nESC-EpiLC-PGCLC in vitro differentiation system. As expected, given their hypomethylated state, most DMS germline genes were not derepressed in *Dnmt* cTKO nESCs. On the other hand, many of these genes were upregulated in nESCs expressing an MGA mutant incapable of binding to E-box sites, or following deletion of *Pcgf6* or *Ring1b*, confirming the critical role that MGA/PRC1.6 plays in silencing of bound germline genes even in the presence of low levels of DNAme (Fig. 6b). Of note however, E-box-mediated binding of MGA/MAX at a subset of germline genes is dispensable for the persistence of H2AK119ub1, indicating that

another pathway may promote recruitment of PRC1.6. While it is tempting to speculate that E2F6 plays a role, KD of this gene in an independently generated *Mga*-ΔHLH ESC line[48] yielded only a modest reduction in PCGF6 binding at GGD genes relative to *Mga*-ΔHLH alone (Supplementary Fig. 6c), indicating that E2F6 may not be the sole factor contributing to PRC1.6 recruitment at these genes in the absence of MGA/MAX binding.

Consistent with the global increase in DNAme in epiblast, most of these germline genes showed a dramatic increase in DNAme in WT EpiLCs, and a subset was derepressed in *Dnmt* cTKO EpiLCs. This includes several GGD genes that are also upregulated in both nESCs and EpiLC following deletion of PRC1.6 subunits, indicating that both DNAme and H2AK119ub1 may be simultaneously engaged in silencing of germline genes in some developmental contexts, such as in epiblast cells. As germline genes are also upregulated in adult liver and brain in *Pcgf6* KO mice[34], DNAme-dependent silencing clearly does not always supersede PRC1.6-mediated silencing following gastrulation. Additional studies will be required to determine the relative importance of these epigenetic marks in silencing of germline genes in other somatic tissues.

We show that de novo DNAme of the CGI promoters of several germline genes in EpiLCs is dependent on the MGA bHLHZ domain (Fig. 6). This observation is consistent with a recent report demonstrating that a few dozen germline genes, including *Ddx4* and *Dazl*, show reduced DNAme following *Max* KD in pESCs[29]. Thus, the establishment of DNAme at the peri-implantation stage may depend upon MGA/MAX binding to the E-box motif and subsequent recruitment of PRC1.6 and SETDB1. Indeed, the promoter regions of most GGD genes also show at least a modest reduction of DNAme in *Ring1b* or *L3mbtl2* KO E6.5 epiblast[67], albeit to a lesser extent than observed in *Mga*-ΔHLH EpiLCs. As DNAme of the E-box motif inhibits MGA/MAX binding[35–37], active DNA demethylation may be essential for MGA/MAX recruitment in cell types expressing relatively high levels of the de novo DNMTs. Conversely, the accumulation of DNAme in somatic tissues over the E-box motifs within the CGI promoter regions of germline genes may preclude PRC1.6 binding, leaving DNAme as the predominant repressive chromatin mark.

As the promoter regions of many germline genes that lack E-box or E2F motifs are efficiently de novo methylated, the presence of H2AK119ub1 and H3K9me3 is clearly not a prerequisite for the establishment of DNAme. Indeed, we find that the subset of DMS germline genes bound by MGA/MAX and E2F6 actually show delayed DNAme both in vitro and in vivo relative to the DMS germline genes not enriched for PRC1.6. While H2AK119ub1[79] and/or SETDB1[80] may play a direct role in recruitment of the de novo DNMTs, inhibition/removal of H3K4 trimethylation is likely a universal prerequisite for de novo DNAme of DMS germline gene promoters[81–83]. As H3K36 di- and tri-methylation also promote de novo DNAme in intergenic and intragenic regions, respectively[84–86], methylation of this H3 residue may also play a role at germline genes.

Recent reports reveal that H2AK119ub1 is deposited in 2-cell stage embryos as early as E1.5, with enrichment of H3K27me3 only clearly evident at most Polycomb targets, in E3.5 ICM[64,65]. The germline genes studied here show similar kinetics of accumulation of these marks, indicating that PRC1.6, like other non-canonical PRC1 complexes, is likely bound to its target regions very early in embryonic development. Deposition of these and other repressive marks at this stage may compensate for the permissive chromatin state associated with the global DNA demethylation that follows fertilization[6]. Embryonic acquisition of H2AK119ub1 is also observed in zebrafish embryos just before ZGA[87], indicating that early deposition of this mark may be

conserved across species. While PRC1.6 and H2AK119ub1 play a critical role in silencing of MGA/MAX-bound germline genes in pESCs, deletion of *Eed* had no effect. As this PRC2 subunit is essential for H3K27me3 deposition, H2AK119ub1 is clearly the more important Polycomb mark for silencing of these loci. This observation is consistent with a recent study showing that deletion of *Eed* in pESCs abolishes H3K27me3 deposition but fails to recapitulate the transcriptional defects observed in the absence of RING1A/B[14].

Although the roles of MGA/MAX and/or E2F6 in the recruitment of PRC1.6 or SETDB1 during early embryogenesis has not been established, mice lacking the gene encoding PRC1.6 subunits or *Setdb1* exhibit early embryonic lethality[31,41,43,45,46,88], which may be due to the precocious expression of specific genes, including DMS germline genes. Of note, Polycomb-dependent repression of germline genes is also observed in species that do not utilize CpG methylation. For example, mutations in *l(3)mbt*, a homolog of *L3mbtl2*, result in ectopic expression of germline genes in larval brain in *D. melanogaster*[89]. Similarly, LIN-61/L3MBTL2 plays a role in silencing of germline genes in somatic cells in *Caenorhabditis elegans*[90]. Notably, MET-2, the SETDB1 homolog in *C. elegans*, is also required for repression of germline genes in somatic tissues, via deposition of H3K9me2 (rather than H3K9me3) at their promoters[91]. Thus, PcG proteins and H3K9 methylation may play an ancestral role in silencing of germline genes in metazoans, with DNAme engaged in the process in mammals only after implantation. As we found only a modest effect of G9a depletion on silencing or de novo DNAme of GGD genes, SETDB1-dependent H3K9me3 deposition is apparently more important for repression of promoter regions harboring E-box and/or E2F motifs than G9a-mediated H3K9me2. This is consistent with a previous report showing dramatically higher expression of several GGD genes in *Pcgf6* KO than *G9a* KO ESCs[34].

Here, we show that PCGF6 is required for H3K9me3 deposition at DMS germline genes in nESCs and EpiLCs. Consistent with this observation, mass spectrometry analyses revealed that PCGF6 interacts with SETDB1 in pESCs[92] and with H3K9me3 in several mouse tissues[59]. However, as deletion of *Pcgf6* in ESCs also compromises MAX as well as RING1B chromatin loading[40], PCGF6 may play an indirect role in SETDB1 recruitment, via stabilization of the PRC1.6 complex at its target sites (Fig. 6b). Destabilization of PRC1.6 following deletion of PCGF6 likely explains why the fold-change in expression of MGA/MAX-bound germline genes in *Pcgf6* KO EpiLCs is similar to that observed following *Setdb1/Ring1b* DKO, as deletion of *Ring1b* does not have the same destabilizing effect[40]. Consistent with a direct role for MGA/MAX-mediated SETDB1 recruitment, MAX was recently shown to interact with SETDB1 via co-immunoprecipitation, and *Ddx4* and *Dazl* promoters showed reduced H3K9me3 in *Max* KD pESCs[29]. However, a recent study suggests that SETDB1 may indirectly interact with PRC1.6, via its known binding partner ATF7IP[93]. ATF7IP contains a C-terminal fibronectin type-III (FNIII) domain, which was shown to interact with MGA in pESCs via mass spectrometry. This mechanism of SETDB1 recruitment would explain why TRIM28 is dispensable for SETDB1-mediated silencing of DMS germline genes, while ATF7IP is required for their silencing[93]. Regardless, our analyses reveal that MGA/MAX-bound GGD genes are generally upregulated in *Setdb1* KO nESC and EpiLCs to a lesser extent than observed in the PRC1.6 subunit mutants, indicating that H3K9me3 plays an important but ancillary role in their silencing.

In summary, we show that H2AK119ub1 and H3K9me3 mark the hypomethylated CGI promoter regions of PRC1.6-bound DMS germline genes during pre-implantation stages, with de novo DNAme occurring at these regions only after implantation.

Consistent with their kinetics in vivo, PRC1.6 and to a lesser extent SETDB1 play important roles in silencing of these genes in nESCs and EpiLCs, suggesting that H2AK119ub1 and H3K9me3 regulate DMS germline genes during pre-implantation embryonic development, prior to the establishment of DNAme. The atypical combination of these histone PTMs, which generally mark distinct genomic regions, and DNAme, likely serves to prevent precocious expression of germline genes during early embryonic development, in the early germline and in somatic tissues, where expression of these genes would otherwise be deleterious.

## Methods

**Mice**. The *Oct4::EGFP* transgenic mice[94] were maintained on a C57BL/6 genetic background. Same-sex littermates were housed in groups of two to five mice per cage with nesting material and a plastic igloo. The mouse facility was kept between 20 and 26 °C and 40% and 60% relative humidity with a 12 h light and dark cycle. Mouse experiments were approved by the Animal Care Committee at the University of British Columbia (UBC) under certificate numbers A16-0230, A16-0269, A20-0229, and A20-0230, with the guidelines from the national Canadian Council on Animal Care (CCAC).

**Isolation of PGCs**. Embryos were obtained from the mating of WT C57BL/6 mice with C57BL/6 mice carrying an *Oct4::EGFP* transgene at embryonic day listed (noon of the day when a copulation plug was identified was designated as embryonic day 0.5 [E0.5]). E9.5 embryos were collected and dissected in DMEM supplemented with 10% fetal bovine serum (FBS). The hindgut endoderm regions that contained PGCs were dissected from embryos and tissue fragments containing PGCs were incubated at 37 °C for 5–8 min in 0.25% trypsin/0.5 mM EDTA/PBS with serial pipetting to obtain single-cell suspensions. After quenching the reaction with FBS, cells were washed once with FACS buffer (0.1% BSA/PBS), counter-stained with propidium iodide (PI) and passed through a 35 μm cell strainer (Falcon). FACS was carried out on a BD Influx (Supplementary Fig. 3d). Following removal of doublet/triplet cells, OCT4::EGFP+ (PI−) cells were sorted into a 1.5 ml low-retention tube containing 500 μl of FACS buffer. Following centrifugation, the supernatant was removed, and the cells were snap-frozen in liquid nitrogen and kept at −80 °C until use.

**Mouse ESC lines**. ESC lines were generated from respective blastocyst embryos as described previously[95]. A published *Ring1b* conditional allele[96] was used to establish *Ring1b* cKO ESCs. This *Ring1b* conditional allele and a published *Setdb1* conditional allele[25] were used to establish *Setdb1/Ring1b* cDKO ESCs. EBRTcH3 ESCs (WT)[97] and their derivative *Mga-ΔHLH* ESCs[74], *Setdb1* cKO ESCs[25], *Pcgf6* cKO ESCs[31] and *Dnmt1/3a/3b* cTKO ESCs[73] were described previously. Male ESCs were used in this study. All cKO ESC lines express the Cre-estrogen receptor fusion protein Cre-ERT2 or MER-Cre-MER, and the addition of 4-OHT induces the deletion of each targeted gene.

**Cell culture**. EpiLC-PGCLC induction was performed essentially as described previously[66] with minor modifications. Briefly, pESCs were adapted to 2i (1 μM of PD0325901, 3 μM of CHIR99021) + LIF feeder-free culture conditions to transit to nESCs. EpiLCs were induced by plating nESCs in a well coated with human plasma fibronectin (Millipore) in N2B27 medium containing Activin A (R&D), bFGF (Gibco) and KnockOut Serum Replacement (KSR; Gibco). For cKO cells, 4OHT (Sigma, dissolved in ethanol) at a final concentration of 100 nM was added to culture medium to induce Cre recombinase activity. Western blotting and/or RT-qPCR were performed to demonstrate depletion of the targeted genes in each cKO cell line upon treatment with 4-OHT for 48 h (Supplementary Fig. 5b, c), as described below. The extended culture of EpiLCs (exEpiLCs) was carried out in accordance with a previous report[98]. PGCLCs were induced under floating aggregation culture conditions by plating EpiLCs in a well of a low-cell-binding U-bottom 96-well plate (Nunc) in GMEM-based serum-free medium in the presence of the cytokines BMP4 (R&D), LIF (ESGRO; Chemicon), SCF (R&D) and EGF (R&D). Cell aggregates including d4 PGCLCs were dissociated with TrypLE (10 min, 37 °C) and incubated with anti-SSEA1 antibody (BioLegend) and anti-CD61 antibody (BioLegend) conjugated with Phycoerythrin (PE) and Alexa Fluor 647, respectively. Cells were washed with FACS buffer and passed through a 35 μm cell strainer. FACS was carried out on a BD Influx (Supplementary Fig. 3d). Following removal of doublet/triplet cells, double-positive PGCLCs were sorted into a 1.5 ml low-retention tube containing 500 μl of FACS buffer. Following centrifugation, the supernatant was removed, and the cells were snap-frozen in liquid nitrogen and kept at −80 °C until use. *Ring1b* cKO pESCs were also cultured in DMEM (Kohjin-bio) with 20% FBS (Sigma), MEM nonessential amino acids, L-glutamine, 2-mercaptoethanol, and LIF on mitomycin C-treated (Sigma) primary MEF feeder layers. For cKO, 4OHT was added to the medium to a final concentration of 800 nM.

**RT-qPCR analysis**. After extraction and purification, total RNA was reverse transcribed using Superscript IV (Thermo Fisher Scientific), and cDNA was used for quantitative PCR with the PowerUP SYBR Green PCR Master Mix (Applied Biosystems). PCR signal was detected by QuantStudio 3 (Applied Biosystems). The sequences of the PCR primers used in this study are listed in Supplementary Data 5.

**Western blotting**. Cells were lysed with modified RIPA buffer (50 mM Tris pH 8.0, 500 mM NaCl, 1% Triton-X, 0.5% Sodium deoxycholate, and 0.1% SDS) supplemented with complete proteinase inhibitor (Roche); lysates were subjected to sonication and clarified by centrifugation. Protein concentrations were determined by Qubit fluorometer (Invitrogen). Protein extracts were separated on 8–10% SDS-PAGE gels, transferred to PVDF membranes, blocked with 5% milk in Tris-buffered saline (TBS: 20 mM Tris–HCl pH 7.5, 100 mM NaCl) or with Intercept (TBS) blocking buffer (LI-COR) and probed overnight at 4 °C with primary antibodies diluted in TBST (TBS plus 0.1% Tween-20) containing 1% milk or in Intercept (TBS) blocking buffer with 0.1% Tween-20: 1:500 rabbit anti-SETDB1 (PtoteinTech 11231-1-AP), 1:500 rabbit anti-PCGF6 (ProteinTech 24103-1-AP), 1:500 rabbit anti-RING1B (Cell Signaling 5694) and 1:2500–5000 mouse anti-ACTB (Invitrogen MA5-15739). Blots were subsequently washed in TBS-T, incubated in IRDye-conjugated secondary antibodies diluted 1:15,000–20,000 in 1% milk in TBS-T or in Intercept (TBS) blocking buffer with 0.1% Tween-20 and 0.01% SDS, washed again in TBS-T and scanned on the ChemiDoc infrared imaging system (BioRad).

**Strand-specific total RNA-sequencing and transcriptome analysis**. Total RNA was extracted from ten thousand nESCs or EpiLCs using the RNeasy Micro kit (Qiagen) with on-column DNaseI treatment. rRNA was depleted using the NEBNext rRNA Depletion Kit (NEB) and cDNA was synthesized using the NEBNext UltraII RNA First Strand Synthesis Module (NEB) and NEBNext UltraII Directional RNA Second Strand Synthesis Module (NEB). Following purification of double-stranded DNA by AMPure XP beads, libraries were constructed as described previously[65], with minor modifications. Briefly, end repair and A-tailing were performed using NEBNext End Repair/dA-Tailing Module (NEB) and ligation was carried out using NEBNext Ultra II Ligation Module (NEB) with an in-house adapter which is compatible with Illumina sequencers. The sequences of the adapter oligo are listed in Supplementary Data 5. After purifying adapter-ligated products by AMPure XP beads (Beckman Coulter), libraries were amplified with unique indexing primers for 12 cycles. Following purification of the amplified libraries by AMPure XP beads, libraries were pooled and run on a 2% E-Gel EX Gel (ThermoFisher Scientific). Fragments from 200–600 bp were excised and purified by Zymoclean Gel DNA Recovery Kit (Zymo Research). The concentration of pooled libraries was determined by qPCR using NEBNext Library Quant Kit for Illumina (NEB), adjusted to 4 nM and then sequenced on the NextSeq 500 platform with 76 paired-end high-output mode according to the manufacturer's instructions. Sequenced reads were then aligned to the mouse genome (mm10) and transcripts aligned using STAR[99] with --sjdbGTFfile and --twopassMode Basic options and then read counts and RPKM (reads per kilobase mapped reads) values for each of the genes were calculated using the RSEM program with default parameter[100]. For analyses of differentially expressed genes (DEGs) between WT and mutant lines, we used TCC (Tag Count Comparison)-GUI, a robust DEG normalization program (https://github.com/swsoyee/TCC-GUI)[76] with default parameters (number of iteration = 3, FDR cut-off = 0.1, elimination of potential DEGs = 0.05) using DESeq2 estimation[101]. Gene ontology analyses were performed with ShinyGO, a graphical gene-set enrichment tool (http://bioinformatics.sdstate.edu/go/)[102]. For *Ring1b* cKO vs WT pESCs, total RNA was prepared from cultured cells using the Allprep DNA/RNA mini kit (Qiagen) following vendor's instructions. Each RNA-seq library was prepared from 1 μg of total RNA, using the NEBNext Ultra Directional RNA Library Prep Kit for Illumina (NEB), following the supplier's instructions. Libraries were quantified with the KAPA Library Quantification Kit (Illumina). Samples were multiplexed and adjusted to 2 nM concentration, followed by Illumina HiSeq2000 sequencing. Reads were generated by 50 bp single-end sequencing.

**Ultra-low-input native chromatin immunoprecipitation (ULI-N-ChIP)-sequencing**. The ULI-N-ChIP-seq protocol[65] was used to generate ChIP-seq libraries, with minor modifications. Briefly, chromatin was fragmented with 3.33 units of Micrococcal Nuclease (NEB), diluted in native ChIP buffer (20 mM Tris-HCl pH 8.0, 2 mM EDTA; 150 mM NaCl, and 0.1% Triton X-100) containing 1 mM PMSF and EDTA-free protease inhibitor cocktail (Roche) and sonicated three times with 5-second cycles on low power (Diagenode Bioruptor). Chromatin fragments were then split into two aliquots (~1000 cells/ChIP) and incubated with each of the histone methylation antibodies at 4 °C overnight. Antibodies used for ChIP experiments were H3K9me3 (39161, lot 13509002, Active Motif, 125 ng) and H2AK119ub1 (8240, lot 6, Cell Signaling, 500 ng). Following purification of ChIPed DNA by AMPure XP beads, libraries were constructed, pooled and sequenced as described above. Libraries were sequenced either on the NextSeq 500 platform with 76 paired-end reads, high-output mode (as for RNA-seq) or on the NextSeq 2000 platform with 66 paired-end reads, P2-output mode. ChIP-seq reads were aligned to the mouse genome (mm10) using bwa[103]. PCR duplicates were removed by Picard-tools (http://broadinstitute.github.io/picard) and reads with low

mapping quality (MAPQ < 5) were excluded by Samtools (http://www.htslib.org/). Processed reads were then used for calculating RPKM values by VisR[104].

**Bisulfite sequencing**. Using the EZ DNA Methylation-Direct Kit (Zymo Research), bisulfite reactions were performed directly on ten thousand EpiLCs or exEpiLCs without DNA extraction. PCR primers were designed with MethPrimer (http://www.urogene.org/methprimer/index1.html)[105] and the primer sequences are listed in Supplementary Data 5. The target regions were amplified by PCR with the EpiTaq HS DNA Polymerase (Takara) using the following conditions: after the first denaturation (3 min at 95 °C), 40 cycles of 30 s at 94 °C, 30 s at 60 °C, 60 s at 72 °C followed by 10 min at 72 °C. The PCR products were cloned by TA cloning in the pGEM T vector (Promega) and 10–16 clones were sequenced. Sequence data were analyzed with the QUantification tool for Methylation Analysis (QUMA; http://quma.cdb.riken.jp/top/index.html)[106].

**Analysis of publicly available data**. The sra format file downloaded from the Sequence Read Archive (SRA) was converted into fastq file format using sratoolkit (https://github.com/ncbi/sra-tools). Read quality was then assessed by FastQC and the reads were processed according to the experimental application, as described above.

**Visualization**. Bedtools was used to generate bedgraph data from bam files. BedGraphToBigWig (UCSC) was used to convert bedgraph data to bigwig format for visualization. Graphs were generated by VisR[104] or Microsoft Excel. Genome browser screenshots were generated using IGV[107] and Venn diagrams were generated using DeepVenn (https://www.deepvenn.com).

**Statistics**. Two-tailed *t*-tests of two samples assuming unequal variances were performed when comparing levels of H3K9me3, H2AK119ub1, or DNAme at DMS germline genes (including GGD genes) or their expression levels between mutant and WT samples. Two-tailed Mann-Whitney *U*-tests were performed between mutant and WT when comparing DNAme levels at the CGI promoters of GGD genes. Violin plots show the median (line inside the Violin), where 50% of the data are distributed.

**Reporting summary**. Further information on research design is available in the Nature Research Reporting Summary linked to this article.

## Data availability
The statistics of datasets generated in this study are summarized in Supplementary Data 6. ChIP-seq and RNA-seq data have been deposited in the Gene Expression Omnibus database under accession number GSE171695. All publicly available data are listed in Supplementary Data 2. These datasets were reprocessed for consistency of data analyses. Source data are provided with this paper.

## Code availability
Publicly available software and VisR[104] were used for analysis and graphical representation of data.

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

## Acknowledgements

We thank J. Brind'Amour, J. Richard Albert, and K. Jensen for helpful discussions and technical assistance; L. Lefebvre (UBC) for critical reading of the manuscript; and T. Stach, R. Vander Werff, Y. Chung, and Y. Zhao (BRC-seq, UBC) for deep sequencing and J. Wong and A. Johnson (ubcFLOW, UBC) for cell sorting. M.C.L. was supported by CIHR grants PJT-153049 and PJT-166170, and NSERC Discovery Grants RGPIN-2021-02808. K.M. was a recipient of a Uehara Memorial Foundation postdoctoral fellowship, a Nakatani Foundation technology exchange grant, a MEXT Fund for the Promotion of Joint International Research 17KK0185, a Mother & Child Health Foundation grant R01-1, and a Sumitomo Foundation grant 200304. This work was also supported by MEXT KAKENHI JP40225446 to J.S. and H.K.

## Author contributions

K.M. and M.C.L. conceived the experiments and wrote the paper with the help of K.S. and A.B.B.; J.S. and H.K. generated the *Dnmt* cTKO, *Pcgf6-*, *Ring1b* cKO, and *Setdb1/Ring1b* cDKO ESC lines and performed RNA-seq in *Ring1b* cKO pESCs. S.M.J. performed the western blot for PCGF6 and RING1B. K.U., A.S., and A.O. generated the *Mga*-ΔHLH ESC line. K.M. performed all other experiments and analyses.

## Competing interests

The authors declare no competing interests.
