## [Peer Review File. · Nature Communications]

Title: Repression of germline genes by PRC1.6 and SETDB1 in the early embryo precedes DNA methylation-mediated silencingREVIEWER COMMENTS

Reviewer #1 (Remarks to the Author):

In the manuscript, Kentaro Mochizuki et al. dissected the functions of DNAm, PRC1.6-mediated histone H2AK119ub1, and H3K9me3 in silencing the germline and GGD genes during early ES cell differentiation or embryo development. The authors showed that in pre-implantation embryos and naïve ESCs promoters of some germline genes and 8 GGD genes are bound by MGA, E2F and enriched with both H2AK119ub1 and H3K9me3. Accordingly, silencing of these genes in nESCs shows a greater dependence on PRC1.6 than DNAm. In contrast, GGD genes are hypermethylated in epiblast-like cells and their silencing is dependent upon SETDB1, PRC1.6/RING1B and DNAm, with H3K9me3 and DNAm establishment dependent upon MGA binding. Thus, GGD genes are initially repressed by PRC1.6, with DNAm subsequently engaged in post-implantation embryos.

Comments:

1. Figure 1 and S1: In the pESCs, there are 55 germline genes have high DNAm and their promoters contain both E2F and E-box motifs (fig. S1D). However, very few genes including 8 GGD gene promoters have MGA, E2F6, Ring1b, Setdb1 binding and enriched with H3K9me3 and H2a119ub1, suggesting that Setdb1 and PRC1.6 recruitment to the GGD gene promoters are not dependent on the E2F and E-box motifs at promoters only. The authors should provide alternative explanation in the discussion section.
2. Figure 3C. There is a discrepancy of expression between Epi in vivo and EpiL in vitro. The authors should discuss this discrepancy.
3. Figure 4. Instead of TET-mediated hydromethylation, the authors did not mention that the low expression of UHRF1 and impaired DNA methylation maintenance are the main reasons leading to DNA hypomethylation in naïve ESCs.
4. Figure 5. In order to determine the roles of MGA, SETDB1, and PRC1.6 in silencing germline genes in nESCs and EpiLCs, the authors used the MGA mutant, Setdb1 and pcgf6-cKO lines. Before draw any further conclusions, the authors should confirm that 2-day 4OHT treatment is sufficient to cause the reduction of SETDB1 and PCGF6 at the protein level.

In summary, the study dissected the role of three major gene silencing complexes in repressing the germline and GGD genes during ESC differentiation and early embryo development, the conclusion was largely data supported. However, based on the previous known knowledge of dynamic DNA methylation in nESC, pESC, pre- and post-implanted embryos and the role of PRC1.6 in repressing germline genes (The polycomb group protein PCGF6 mediates germline gene silencing by recruiting histone-modifying proteins to target gene promoters, JBC, 2020), the results are largely expected and lack of exciting novel findings.

Reviewer #2 (Remarks to the Author):

This manuscript by Mochizuki and colleagues investigates the hierarchy of epigenetic silencing that regulates germline-specific genes. Analogously to imprinted genes, this subset of genes are a paradigm for epigenetic regulation, with the confluence of polycomb, facultative heterochromatin and DNA methylation intersecting to ensure their silencing across somatic tissues and during development. As such, deciphering how these mechanisms interact is of interest both to advance our mechanistic understanding and as a general principle for epigenetic control of transcription. The authors make extensive use of extant datasets and add new data to come to the conclusion that PRC1.6 and H2AK119ub1 operates developmentally upstream of DNA methylation to ensure silencing during naïve pluripotent phases. In later postimplantation/differentiated cells a combination of DNA methylation/SETDB1/PRC1.6 coordinate silencing across distinct germline gene loci. This is dependent on MGA/E2F6 targeting. The manuscript is well-written and clear.

The data throughout the manuscript is convincing, and the conclusions fully warranted. Nevertheless, a major part of the confidence in the conclusions is that many (most) of the principles being presented are already the paradigm in the field and thus the advance in understanding is relatively nuanced. For example, working backwards, it has long been established the germline genes are regulated by promoter DNA methylation in somatic/postimplantation cells (Borgel et al, 2010; Maatouk et al, 2006; Hackett et al, 2012; Weber et al, 2007). It is also clear that H3K27me and/or H3K9me operates at germline genes prior to this during early development (Karimi et al, 2011; Kurimoto et al, 2015), and that PRC1.6 is directly involved (Liu et al, 2020; Stielow et al, 2018), and finally that E2F6/MGA are key components that target these systems (Velasco et al, 2010). Thus I find the data presented to be entirely solid and well constructed, and therefore have no major reservations on the conclusions. However, I am somewhat skeptical that there is any more than an incremental advance in our understanding based on the existing literature. Whilst the authors do make use of published in vivo data, thereby confirming regulatory principles in an embryo, this closely reflects the well-charactered events from in vitro models. It is fair to say that the authors acknowledge existing studieswork, but nevertheless spend the majority of the figures re-confirming these results. In my opinion whilst representing a fair and well-performed manuscript, it may lack novelty.

Borgel J, Guibert S, Li Y, Chiba H, Schubeler D, Sasaki H, Forne T, Weber M (2010) Targets and dynamics of promoter DNA methylation during early mouse development. *Nat Genet* 42: 1093-1100

Hackett JA, Reddington JP, Nestor CE, Dunican DS, Branco MR, Reichmann J, Reik W, Surani MA, Adams IR, Meehan RR (2012) Promoter DNA methylation couples genome-defence mechanisms to epigenetic reprogramming in the mouse germline. *Development* 139: 3623-3632

Karimi MM, Goyal P, Maksakova IA, Bilenky M, Leung D, Tang JX, Shinkai Y, Mager DL, Jones S, Hirst M et al (2011) DNA methylation and SETDB1/H3K9me3 regulate predominantly distinct sets of genes, retroelements, and chimeric transcripts in mESCs. *Cell Stem Cell* 8: 676-687

Kurimoto K, Yabuta Y, Hayashi K, Ohta H, Kiyonari H, Mitani T, Moritoki Y, Kohri K, Kimura H, Yamamoto T et al (2015) Quantitative Dynamics of Chromatin Remodeling during Germ Cell Specification from

Mouse Embryonic Stem Cells. *Cell Stem Cell* 16: 517-532

Liu M, Zhu Y, Xing F, Liu S, Xia Y, Jiang Q, Qin J (2020) The polycomb group protein PCGF6 mediates germline gene silencing by recruiting histone-modifying proteins to target gene promoters. *J Biol Chem* 295: 9712-9724

Maatouk DM, Kellam LD, Mann MR, Lei H, Li E, Bartolomei MS, Resnick JL (2006) DNA methylation is a primary mechanism for silencing postmigratory primordial germ cell genes in both germ cell and somatic cell lineages. *Development* 133: 3411-3418

Stielow B, Finkernagel F, Stiewe T, Nist A, Suske G (2018) MGA, L3MBTL2 and E2F6 determine genomic binding of the non-canonical Polycomb repressive complex PRC1.6. *PLoS Genet* 14: e1007193

Velasco G, Hube F, Rollin J, Neuillet D, Philippe C, Bouzinba-Segard H, Galvani A, Viegas-Pequignot E, Francastel C (2010) Dnmt3b recruitment through E2F6 transcriptional repressor mediates germ-line gene silencing in murine somatic tissues. *Proceedings of the National Academy of Sciences* 107: 9281-9286

Weber M, Hellmann I, Stadler MB, Ramos L, Paabo S, Rebhan M, Schubeler D (2007) Distribution, silencing potential and evolutionary impact of promoter DNA methylation in the human genome. *Nat Genet* 39: 457-466

Reviewer #3 (Remarks to the Author):

The majority of protein coding gene promoters are protected from DNA methylation-based regulation, presumably because DNA methylation confers stable silencing that would be difficult to overcome upon a developmental or environmental cue. A notable exception are germline-specific genes; this class of genes are frequently repressed via DNA methylation in somatic tissues, however during germline epigenetic reprogramming in mammals, the methyl-mark is erased allowing for expression during germline progression. This class of genes presents a fascinating case study in DNA methylation biology for two reasons: 1) understanding the basis of their repression during embryonic methylation reprogramming (which precedes the germline program) when they are also devoid of DNA methylation and 2) determining how de novo DNA methyltransferase target these specific genes and not others. In this impressive study, Mochizuki et al set out to address these questions taking advantage of a bevy of mouse in vivo and in cellula published data sets to make a coherent step-wise regulatory analysis of DNA methylation sensitive genes, with an emphasis on so-called “germline genome defence” (GGD) genes, which are important for transposon repression in the germline. Gaps in the data are filled in with original experiments. Indeed, the regulation of this class of genes is quite intricate. Consistent with previous studies, here they show that MAX/MGA—two proteins associated with non-canonical PRC1.6—target several germline genes through the E-Box binding domain. As such, these genes are enriched for H2AK119ub. Curiously, PRC2 does not play a major role in regulation of these genes, at least in serum-grown ESCs. Instead, MGA recruits SETDB1, which deposits H3K9me3. In other words, two chromatin pathways typically not thought to co-regulate repression cooperate at this class of genes. During

differentiation, DNA methylation eventually plays the more prominent silencing role, which would then persist throughout somatic differentiation. These layers of repression are then stripped away upon germline specification.

Overall, I found this to be an important study for both the fields of epigenetics and mammalian developmental biology. Also, I appreciate that the authors made use of such a wealth of publicly available data from a variety of publications, and were able to implement them to create a rich study. This should be applauded. On the other hand, different labs may perform experiments in slightly different manners, and certainly in the case of in vitro studies where culture artifacts are common, not being in full control of the protocols may lead to a degree of uncertainty when comparing datasets. Nevertheless, overall I am very positive about the work submitted here. I suggest the following improvements for a revised version:

Major comments

1. One glaring omission in this study is the lack of mention of G9a/GLP. These H3K9 methyltransferases deposit H3K9me2 (not H3K9me3). Previous studies demonstrated that they interact with the protein L3MBTL2, a subunit of PRC1.6 (PMID: 22770845). Perhaps more compellingly, a previous in vivo study showed that in G9a mutants (also called EHMT2), DNA methylation is impacted at a number of germline-specific genes (PMID: 26576615). Hence, I was confused why this aspect of germline regulation was never discussed. I suggest for the next version of the paper, the available data from the studies cited above (and perhaps other relevant ones) are incorporated into their model. Or if there are compelling reasons for why these data should not be included, this should be discussed.
2. An aspect of the paper I found very confusing was on the precise epistatic context of MGA in the control of the GGD genes. In the DNA binding mutant ((Mga- Δ HLA), I was surprised that H2Aub was not strongly impacted given the model wherein MAX/MGA recruit PRC1.6 to chromatin. However, there is a decrease of H3K9me3. To me it would then make sense that the Setdb1 mutation would therefore have a strong effect on transcription control. But in fact, the effect is **less** than that observed in the Mga- Δ HLA mutant. For the first part of this comment, the authors suggest H2Aub accumulates because E2F6 (also a DNA-binding factor part of PRC1.6) is sufficient to recruit the complex. This should be proven with a genetic test, ie an Mga/E2f6 double mutant. Secondly, can the authors discuss if it's not H2Aub nor H3K9me3, why the Mga mutant exhibits the biggest effect on GGD gene expression. What is the means of repression? Is it possible it is through G9a/H3K9me2, as discussed above?
3. The authors did a nice job demonstrating the various impacts of DNA methylation in a suite of mutants. However, I am curious about the crosstalk in the reverse direction. Namely, in EpiLCs where DNA methylation plays an important regulatory role, is H3K9me3 impacted in DNA methylation mutants, or does the mark linger? Given the correlation between DNAm and H3K9me3, I think it is worth doing a ChIP experiment in the Dnmt cKO line. This would have implications for mode of DNA methylation based silencing throughout life in somatic tissues. Related, the recently published H3K9me3 triple KO mouse was recently published (PMID: 30606806). I would be curious to know if germline genes are ectopically

expressed in the tissues they analyzed.

4. One thing that was in the back of my mind was that the 2 day EpiLC differentiation is not sufficient to obtain the levels of DNA methylation observed in the in vivo E6.5 epiblast. I understand the rationale for doing a 2 day differentiation: after 2 days the EpiLCs lose their PGCLC competency. On the other hand, this might not be enough time to properly assess the DNA methylation. For example, in their supplementary figure 6, I was quite surprised by the low levels of DNA methylation at the Mael promoter, when at later stages of differentiation, this region is densely methylated. This may explain why several GGD genes are not impacted at the level of expression in the Dnmt cKO EpiLCs in figure 4E and 5D (including Mael). I would still contend that eventually, DNA methylation plays an important role at several of these genes. It would strengthen the message of the paper to do an extended EpiLC differentiation (4-5 days), and observe by RT-qPCR if other GGD genes exhibit DNA methylation sensitivity. The BS + Sanger could be repeated at Mael and Mov10l1 to show that at this later stage, DNA methylation has accumulated to a greater extent.

Minor point

1. This is semantical, but I do not like the term “primed ESCs” for ESCs grown in serum, as it can be conflated with primed pluripotency observed in EpiSCs or human ESCs, which is not the case. I prefer “serum-grown ESCs” to avoid confusion.

Point-by-point response to the reviewers' comments

Reviewer #1 (Remarks to the Author):

In the manuscript, Kentaro Mochizuki et al. dissected the functions of DNAm, PRC1.6-mediated histone H2AK119ub1, and H3K9me3 in silencing the germline and GGD genes during early ES cell differentiation or embryo development. The authors showed that in pre-implantation embryos and naïve ESCs promoters of some germline genes and 8 GGD genes are bound by MGA, E2F and enriched with both H2AK119ub1 and H3K9me3. Accordingly, silencing of these genes in nESCs shows a greater dependence on PRC1.6 than DNAm. In contrast, GGD genes are hypermethylated in epiblast-like cells and their silencing is dependent upon SETDB1, PRC1.6/RING1B and DNAm, with H3K9me3 and DNAm establishment dependent upon MGA binding. Thus, GGD genes are initially repressed by PRC1.6, with DNAm subsequently engaged in post-implantation embryos.

Comments:

1. Figure 1 and S1: In the pESCs, there are 55 germline genes have high DNAm and their promoters contain both E2F and E-box motifs (fig. S1D). However, very few genes including 8 GGD gene promoters have MGA, E2F6, Ring1b, Setdb1 binding and enriched with H3K9me3 and H2a119ub1, suggesting that Setdb1 and PRC1.6 recruitment to the GGD gene promoters are not dependent on the E2F and E-box motifs at promoters only. The authors should provide alternative explanation in the discussion section.

We thank the reviewer for this observation. We have now intersected the 55 DMS germline gene promoters which have high DNAm and both E-box and E2F motifs, with ChIP-seq data for MGA and E2F6 as well as SETDB1, RING1B, H3K9me3 and H2AK119ub1. This analysis reveals that of the 55 genes, 21, including 7/8 GGD genes, show bona fide binding of MGA and E2F6 and enrichment of SETDB1, RING1B, H3K9me3 and H2AK119ub1. Notably, we find that some MGA or E2F6 peaks overlap with variants of E-box/E2F motifs, such as CGCGTG instead of CACGTG or TCCCAC instead of TCCCGC, likely explaining why a number of genes bound by MGA and/or E2F6 (based on ChIPseq) do not include the relevant canonical motifs in their promoter regions.

This analysis is shown below & included in the revised manuscript as a new Supplementary Fig. 1c:

2. Figure 3C. There is a discrepancy of expression between Epi in vivo and EpiL in vitro. The authors should discuss this discrepancy.

We thank the reviewer for this comment. In the original EpiLC/PGCLC paper, Hayashi et al. (2011) mention that the EpiLC transcriptome is very similar to that of epiblast at E5.75 (microarray data). We, therefore, compared RNA-seq data from EpiLCs with those from E5.5 or E6.5 epiblasts and

confirmed that EpiLC are closer to E5.5 than E6.5 epiblast. This is consistent with the observation that the expression of DMS germline genes, including some GGD genes, is transiently upregulated in E5.5 epiblast cells as well as EpiLCs (Fig. 3c). Unfortunately, there are few ChIP-seq or WGBS datasets from E5.5 epiblast. Thus, we chose to present data for E6.5 epiblast in the manuscript.

Relationship between expression in EpiLCs and E5.5 versus E6.5 epiblast:

Hayashi et al., *Cell*, 2011.

Fig. 3c

3. Figure 4. Instead of TET-mediated hydromethylation, the authors did not mention that the low expression of UHRF1 and impaired DNA methylation maintenance are the main reasons leading to DNA hypomethylation in naïve ESCs.

We have now edited the sentences in question to clearly state that UHRF1 (as well as DNMT3A and DNMT3B) is expressed at a lower level in nESCs than pESCs. We have also edited the relevant sentence to avoid implying that the low level of DNAm is due to TET activity. The revised sentences are as follows: “While nESCs express lower levels of *Dnmt3a*, *Dnmt3b* and *Uhrf1* than pESCs^{68,69}, they do express TET1, which mediates oxidation of 5-methylcytosine (5mC) to 5-hydroxymethylcytosine (5hmC) in an active DNA demethylation pathway⁷⁰. As bisulphite sequencing cannot discriminate between 5mC and 5hmC, we surmised that the DNAm observed in DMS germline genes in nESCs may reflect at least in part the presence of 5hmC.”

Parentetically, though we do not include this information in the manuscript in the interest of space, the lower levels of UHRF1 activity in ESCs was recently reported to be a downstream consequence of TET1/2 activity acting specifically on the DPPA3 locus (Mulholland, C. B. et al. Recent evolution of a TET-controlled and DPPA3/STELLA-driven pathway of passive DNA demethylation in mammals. *Nature Communications* 11, 5972–24 (2020). The higher level of *Dppa3* expression promoted by TET1/2 leads to sequestration of UHRF1 via a direct interaction with DPPA3, and in turn disrupted maintenance DNAm.

4. Figure 5. In order to determine the roles of MGA, SETDB1, and PRC1.6 in silencing germline genes in nESCs and EpiLCs, the authors used the MGA mutant, *Setdb1* and *pcgf6*-cKO lines.

Before draw any further conclusions, the authors should confirm that 2-day 4OHT treatment is sufficient to cause the reduction of SETDB1 and PCGF6 at the protein level.

We thank the reviewer for this comment. To confirm reduction of SETDB1, PCGF6 and RING1B in each cKO line following two days of 4OHT treatment, we have now added RT-qPCR as well as Western analyses in Supplementary Fig. 5b-c, respectively.

Efficient depletion of SETDB1, PCGF6 and RING1B upon 2-day 4OHT treatment:

Supplementary Fig. 5

In summary, the study dissected the role of three major gene silencing complexes in repressing the germline and GGD genes during ESC differentiation and early embryo development, the conclusion was largely data supported. However, based on the previous known knowledge of dynamic DNA methylation in nESC, pESC, pre- and post-implanted embryos and the role of PRC1.6 in repressing germline genes (The polycomb group protein PCGF6 mediates germline gene silencing by recruiting histone-modifying proteins to target gene promoters, JBC, 2020), the results are largely expected and lack of exciting novel findings.

We thank the reviewer for summarizing our results and confirming that the conclusions drawn are largely supported by the data presented. Regarding the novelty of our findings, the reviewer mentions the study published in JBC by Liu and colleagues on the role of PCGF6 in mediating silencing of germline genes in somatic cells. Our results are indeed consistent with those reported by Liu et al., including the observation that *Pcgf6* deficiency has only a modest effect on DNAm (in our data in mutant EpiLCs, in Liu et al. in mutant liver). However, unique to our study is the analysis of the impact of the MGA-HLH mutant on DNAm levels (ie de novo DNAm) specifically in EpiLCs. This mutant shows much lower levels of DNAm at the *Mael* and *Mov101* gene promoters than the *Pcgf6* mutant, revealing a role for MGA in this process independent of PCGF6. Furthermore, DNAm remains very low in the MGA-HLH mutant even in otherwise highly methylated exEpiLCs (new Supplementary Figure 7a). These results indicate that not all subunits of PRC1.6 are “created equal”, with respect to how they influence de novo DNAm at implantation. In addition, we show that a subset of GGD genes are silenced largely independent of DNAm even in EpiLCs, i.e. after de novo DNAm has commenced. As discussed in the manuscript, this contrasts with a widely cited paper in the literature (Hackett *et al.* Promoter DNA methylation couples genome-defence mechanisms to epigenetic reprogramming in the mouse germline, *Development* **139**, 3623–3632, 2012), that claimed “Using a novel epigenetic disruption and recovery screen and genetic analyses, we identified a core set of germline-specific genes that are dependent exclusively on promoter DNA methylation for initiation and maintenance of developmental silencing”. We feel that the novelty of our manuscript lies in the combinatorial and systematic comparison of the disruption of multiple

chromatin modifiers in the regulation of germline gene DNA methylation, covalent histone modifications and expression, coupled with the kinetics of establishment of these various chromatin marks during early embryonic development.

Reviewer #2 (Remarks to the Author):

This manuscript by Mochizuki and colleagues investigates the hierarchy of epigenetic silencing that regulates germline-specific genes. Analogously to imprinted genes, this subset of genes are a paradigm for epigenetic regulation, with the confluence of polycomb, facultative heterochromatin and DNA methylation intersecting to ensure their silencing across somatic tissues and during development. As such, deciphering how these mechanisms interact is of interest both to advance our mechanistic understanding and as a general principle for epigenetic control of transcription. The authors make extensive use of extant datasets and add new data to come to the conclusion that PRC1.6 and H2AK119ub1 operates developmentally upstream of DNA methylation to ensure silencing during naïve pluripotent phases. In later postimplantation/differentiated cells a combination of DNA methylation/SETDB1/PRC1.6 coordinate silencing across distinct germline gene loci. This is dependent on MGA/E2F6 targeting. The manuscript is well-written and clear.

The data throughout the manuscript is convincing, and the conclusions fully warranted. Nevertheless, a major part of the confidence in the conclusions is that many (most) of the principles being presented are already the paradigm in the field and thus the advance in understanding is relatively nuanced. For example, working backwards, it has long been established the germline genes are regulated by promoter DNA methylation in somatic/postimplantation cells (Borgel et al, 2010; Maatouk et al, 2006; Hackett et al, 2012; Weber et al, 2007). It is also clear that H3K27me and/or H3K9me operates at germline genes prior to this during early development (Karimi et al, 2011; Kurimoto et al, 2015), and that PRC1.6 is directly involved (Liu et al, 2020; Stielow et al, 2018), and finally that E2F6/MGA are key components that target these systems (Velasco et al, 2010). Thus I find the data presented to be entirely solid and well constructed, and therefore have no major reservations on the conclusions. However, I am somewhat skeptical that there is any more than an incremental advance in our understanding based on the existing literature. Whilst the authors do make use of published in vivo data, thereby confirming regulatory principles in an embryo, this closely reflects the well-charactered events from in vitro models. It is fair to say that the authors acknowledge existing studies/work, but nevertheless spend the majority of the figures re-confirming these results. In my opinion whilst representing a fair and well-performed manuscript, it may lack novelty.

We thank the reviewer for summarizing our results and their supportive comments on the data presented and conclusions drawn. Please see our reply to Reviewer 1 concerning our views on the novelty of our findings relative to previously published studies.

Reviewer #3 (Remarks to the Author):

The majority of protein coding gene promoters are protected from DNA methylation-based regulation, presumably because DNA methylation confers stable silencing that would be difficult to overcome upon a developmental or environmental cue. A notable exception are germline-specific genes; this class of genes are frequently repressed via DNA methylation in somatic tissues, however during germline epigenetic reprogramming in mammals, the methyl-mark is erased allowing for expression during germline progression. This class of genes presents a fascinating case study in DNA methylation biology for two reasons: 1) understanding the basis of their repression during

embryonic methylation reprogramming (which precedes the germline program) when they are also devoid of DNA methylation and 2) determining how *de novo* DNA methyltransferase target these specific genes and not others. In this impressive study, Mochizuki et al set out to address these questions taking advantage of a bevy of mouse *in vivo* and *in cellular* published data sets to make a coherent step-wise regulatory analysis of DNA methylation sensitive genes, with an emphasis on so-called “germline genome defence” (GGD) genes, which are important for transposon repression in the germline. Gaps in the data are filled in with original experiments. Indeed, the regulation of this class of genes is quite intricate. Consistent with previous studies, here they show that MAX/MGA—two proteins associated with non-canonical PRC1.6—target several germline genes through the E-Box binding domain. As such, these genes are enriched for H2AK119ub. Curiously, PRC2 does not play a major role in regulation of these genes, at least in serum-grown ESCs. Instead, MGA recruits SETDB1, which deposits H3K9me3. In other words, two chromatin pathways typically not thought to co-regulate repression cooperate at this class of genes. During differentiation, DNA methylation eventually plays the more prominent silencing role, which would then persist throughout somatic differentiation. These layers of repression are then stripped away upon germline specification.

Overall, I found this to be an important study for both the fields of epigenetics and mammalian developmental biology. Also, I appreciate that the authors made use of such a wealth of publicly available data from a variety of publications, and were able to implement them to create a rich study. This should be applauded. On the other hand, different labs may perform experiments in slightly different manners, and certainly in the case of *in vitro* studies where culture artifacts are common, not being in full control of the protocols may lead to a degree of uncertainty when comparing datasets.

Nevertheless, overall I am very positive about the work submitted here.

We thank the reviewer for summarizing our results and their supportive words regarding our analysis of publicly available data from a variety of sources. We also appreciate the reviewer’s comments on the care that must be taken when comparing datasets from different sources.

I suggest the following improvements for a revised version:

Major comments

1. One glaring omission in this study is the lack of mention of G9a/GLP. These H3K9 methyltransferases deposit H3K9me2 (not H3K9me3). Previous studies demonstrated that they interact with the protein L3MBTL2, a subunit of PRC1.6 (PMID: 22770845). Perhaps more compellingly, a previous *in vivo* study showed that in G9a mutants (also called EHMT2), DNA methylation is impacted at a number of germline-specific genes (PMID: 26576615). Hence, I was confused why this aspect of germline regulation was never discussed. I suggest for the next version of the paper, the available data from the studies cited above (and perhaps other relevant ones) are incorporated into their model. Or if there are compelling reasons for why these data should not be included, this should be discussed.

We thank the reviewer for this comment. In Supplementary Fig. 2c, we have now added an intersection between G9A enrichment and the change in gene expression following G9a KO in pESCs. As shown below, we find that while a number of DMS germline genes are upregulated, GGD genes show low or no upregulation. As the reviewer mentions, we have confirmed that DNAm at DMS germline gene promoters is reduced in G9a KO epiblast cells (new Fig. 5f, g). However, DNAm at these regions is not lower in G9a KO E8.5 embryos (new Supplementary Fig. 8b, c). Thus, G9A is apparently required for early *de novo* DNAm following implantation, but this delay in DNAm is compensated for/overcome by E8.5. This observation is consistent with the recent

analysis by Weber and colleagues (Dahlet et al, 2021) in Nature Communications (published after this manuscript was submitted), who found: 1. “no enrichment of HDAC1/2 or G9a/GLP in the E2F6 interactome in ESCs” 2. that almost all E2F6-repressed germline genes are not derepressed in G9a KO ESCs and 3. that E2F6 and G9a repress distinct sets of genes in mouse embryos. In addition, the authors showed that most germline genes repressed by G9a are not bound by E2F6, suggesting that they act in distinct pathways. Taken together, these data indicate that E2F6 and MGA/MAX function independent of G9a. We now also mention this observation in the revised Discussion.

Role of G9A in silencing of germline genes is independent of PRC1.6 and dispensable for DNAmE in their promoters at later stage:

2. An aspect of the paper I found very confusing was on the precise epistatic context of MGA in the control of the GGD genes. In the DNA binding mutant (*Mga-ΔHLH*), I was surprised that H2Aub was not strongly impacted given the model wherein MAX/MGA recruit PRC1.6 to chromatin. However, there is a decrease of H3K9me3. To me it would then make sense that the *Setdb1* mutation would therefore have a strong effect on transcription control. But in fact, the effect is *less* than that observed in the *Mga-ΔHLH* mutant. For the first part of this comment, the authors suggest H2Aub accumulates because E2F6 (also a DNA-binding factor part of PRC1.6) is sufficient to recruit the complex. This should be proven with a genetic test, ie an *Mga/E2f6* double mutant.

Like the reviewer, we were also perplexed by the observation that H2Aub was not more strongly impacted in the *Mga-ΔHLH* mutant. While we did propose that E2F6 may compensate for the loss of MGA/MAX binding (by independently promoting PRC1.6 recruitment), a careful review of the literature reveals that this is unlikely to be the case. Scelfo et al. (Mol Cell, 2019; GSE122715), essentially carried out the experiment suggested by the reviewer, namely depletion of both E2F6 and MGA binding. Our new analysis of the data presented in this study revealed that enrichment of PCGF6 (a core subunit of PRC1.6), is only modestly further reduced at germline gene promoters in *Mga-ΔHLH/E2f6* KD double mutant ESCs relative to control *Mga-ΔHLH* ESCs (see panel below). Importantly, this includes at GGD genes.

Relative to *Mga*-ΔHLH alone, the combination of *Mga*-ΔHLH and *E2f6* KD has little further impact on PRC1.6 recruitment at MAX/MGA bound GGD genes:

We have revised the relevant text in our manuscript as follows: “Of note however, E-box-mediated binding of MGA/MAX at a subset of germline genes is dispensable for the persistence of H2AK119ub1, indicating that another pathway may promote recruitment of PRC1.6. While it is tempting to speculate that E2F6 plays a role, KD of this gene in an independently generated *Mga*-ΔHLH ESC line⁴⁸ yielded only a modest reduction in PCGF6 binding at GGD genes relative to *Mga*-ΔHLH alone (Supplementary Fig. 6c), indicating that E2F6 may not be the sole factor contributing to PRC1.6 recruitment at these genes in the absence of MGA/MAX binding.”

Secondly, can the authors discuss if it's not H2Aub nor H3K9me3, why the *Mga* mutant exhibits the biggest effect on GGD gene expression. What is the means of repression? Is it possible it is through G9a/H3K9me2, as discussed above?

Supplementary Fig. 6c

As mentioned above, GGD genes were minimally upregulated in *G9a* KO pESCs (even though other DMS germline genes were upregulated), suggesting that *G9A* is unlikely to be involved in silencing of these MGA/MAX/PRC1.6 bound genes. Rather, as detailed below, we propose that the transcription factor MEIOSIN may be involved. We have revised the relevant sentences as follows: “Intriguingly, the fold-change in expression of some GGD genes in *Mga*-ΔHLH EpiLCs and exEpiLCs far exceeds that observed in EpiLCs derived from the other KO lines, including *Setdb1* cKO (Fig. 5d and Supplementary Fig. 6a), indicating that MGA/MAX play a particularly important role in repression of GGD gene expression. Notably, the transcription factor MEIOSIN was recently reported to play a critical role in driving expression of germline genes, including GGD genes, in testis⁷⁷. As this gene shows greater upregulation in *Mga*-ΔHLH nESCs/EpiLCs than any of the other KO lines analyzed (Supplementary Fig. 6a), the relatively high level of de-repression of GGD genes in *Mga*-ΔHLH cells may be due to the loss of PRC1.6 and SETDB1 binding combined with increased binding of MEIOSIN at their promoter regions.”

Greater upregulation of GGD genes in *Mga*-ΔHLH nESCs/EpiLCs may be due to loss of PRC1.6/SETDB1 recruitment and concomitant binding of MEIOSIN at their promoter regions:

Supplementary Fig. 6a,b

3. The authors did a nice job demonstrating the various impacts of DNA methylation in a suite of mutants. However, I am curious about the crosstalk in the reverse direction. Namely, in EpiLCs where DNA methylation plays an important regulatory role, is H3K9me3 impacted in DNA methylation mutants, or does the mark linger? Given the correlation between DNAm and H3K9me3, I think it is worth doing a ChIP experiment in the Dnmt cKO line. This would have implications for mode of DNA methylation based silencing throughout life in somatic tissues.

We thank the reviewer for this suggestion. As requested, we now demonstrate that H3K9me3 and H2AK119ub1 in *Dnmt* cTKO nESCs and EpiLCs are only modestly reduced at DMS germline genes, including GGD genes (Fig. 5a, b). These observations suggest that loss of DNAm has only minimal impact on H3K9me3 and H2AK119ub1, at least in these cell types.

H3K9me3 and H2AK119ub1 is slightly impacted in DNAm mutant cells:

Related, the recently published H3K9me3 triple KO mouse was recently published (PMID: 30606806). I would be curious to know if germline genes are ectopically expressed in the tissues they analyzed.

In the manuscript mentioned, “H3K9me3-heterochromatin loss at protein-coding genes enables developmental lineage specification” -published in 2019, Zaret and colleagues generated an endoderm-specific conditional triple-knockout mutant (TKO) mouse strain of the H3K9 KMTases Setdb1, Suv39h1 and Suv39h2, which showed a marked decrease in H3K9me3. They subsequently carried out RNA-seq analysis on liver of 1-month-old TKO mice. The authors reported 476 upregulated genes in the TKO, 25 of which are in the “GO:0007283~spermatogenesis” GOTERM_BP_DIRECT GO category (Table S16). Nevertheless, intersection of this 476 upregulated

genes list with our list of 137 DMS germline genes (Supplementary data 1) yielded only 2 genes in common (Stag3 and Hsf2bp). While Hsf2bp, is among the “GO:0007283~spermatogenesis” genes, neither of these genes are GGD genes. Thus, our germline genes of interest, ie those regulated by PRC1.6 and SETDB1 in the embryo, do not seem to be upregulated in the liver in the absence of

Setdb1/Suv39h1/Suv39h2. This is consistent with the observations of Liu et al. mentioned above (“The polycomb group protein PCGF6 mediates germline gene silencing by recruiting histone-modifying proteins to target gene promoters”, JBC, 2020), in which it was shown that PRC1.6 can still function to repress germline genes in adult somatic cells, including in the liver.

4. One thing that was in the back of my mind was that the 2 day EpiLC differentiation is not sufficient to obtain the levels of DNA methylation observed in the in vivo E6.5 epiblast. I understand the rationale for doing a 2 day differentiation: after 2 days the EpiLCs lose their PGCLC competency. On the other hand, this might not be enough time to properly assess the DNA methylation. For example, in their supplementary figure 6, I was quite surprised by the low levels of DNA methylation at the Mael promoter, when at later stages of differentiation, this region is densely methylated. This may explain why several GGD genes are not impacted at the level of expression in the Dnmt cKO EpiLCs in figure 4E and 5D (including Mael). I would still contend that eventually, DNA methylation plays an important role at several of these genes. It would strengthen the message of the paper to do an extended EpiLC differentiation (4-5 days), and observe by RT-qPCR if other GGD genes exhibit DNA methylation sensitivity. The BS + Sanger could be repeated at Mael and Mov10i1 to show that at this later stage, DNA methylation has accumulated to a greater extent.

We thank the reviewer for this suggestion. Unfortunately, as shown below, we could not induce extended EpiLCs (exEpiLCs, i.e. day 7 post induction of differentiation) from Dnmt cTKO or Dnmt3 cDKO nESCs, due to high levels of cell lethality or abnormal differentiation of surviving cells (see panel below). However, we were able to generate in parallel exEpiLCs from Mga-HLH nESCs.

Therefore, we focused on the impact of absence of E-box binding of MGA on DNAm accumulation at GGD gene promoters as well as on silencing of these genes in exEpiLCs. As the reviewer speculates, we now demonstrate that almost all GGD genes are further upregulated in Mga-ΔHLH exEpiLCs compared to nESCs/EpiLCs (NEW Supplementary Fig. 6b and shown below). We also show that DNAm at the promoters of Mael and Mov10i1, which further accumulates in WT exEpiLCs, is dramatically reduced in this mutant (see new Supplementary Fig. 7a, b, shown below).

Taken together these results clearly reveal that accumulation of DNAm at such GGD genes after prolonged EpiLC culture is dependent upon the MGA HLH domain, likely via binding to the E-box embedded in their promoter region.

Supplementary Fig. 6b

Supplementary Fig. 7

Minor point

1. This is semantical, but I do not like the term “primed ESCs” for ESCs grown in serum, as it can be conflated with primed pluripotency observed in EpiSCs or human ESCs, which is not the case. I prefer “serum-grown ESCs” to avoid confusion.

We thank the reviewer for this concern. We also struggled with this acronym choice. We would prefer to stick with “pESCs”, as the relatively short acronym is useful in figure panels etc., versus the relatively lengthy “serum-grown ESCs”. As we clearly define “pESCs” in the manuscript, we hope that the reviewer will be comfortable with this decision.

REVIEWERS' COMMENTS

Reviewer #1 (Remarks to the Author):

Thank the authors for addressing all my questions. I recommend to have this MS published in NC.

Reviewer #2 (Remarks to the Author):

The authors have failed to adequately address my critical concern that there is highly limited novelty in the manuscript. The vast bulk of results (and conclusions) draws together the existing literature and paradigms; namely that DNA methylation becomes important for GGD silencing during differentiation in conjunction with SETDB1 and PRC1-E2F6-MGA-MAX, whereas naïve pluripotent cells, which are globally hypomethylated, preferentially rely on the latter systems for repression (see multiple references noted in original comments - not addressed). The authors argument that the unique novelty relies on identification of MGA independent of PCGF6 is highly nuanced (to the point it is not even articulated in the abstract) and also broadly established in any case (see recent PMID: 34117224 and prior work). Moreover this actually reflects a minor portion of the data; the great majority of the figures reinforce extant data and concepts. Finally, in naïve cells GGD are repressed but not silenced (expression is detected), thus full silencing as opposed to dampening does seem to be linked with acquisition of DNAm in differentiated cells in most cases (in collaboration with PRC/SETDB1/MGA, as established in several papers). To me the authors data does not seem to contrast with the central conclusions of PMID: 22949617 that DNAm is critical in differentiated tissues (EpiLC are not differentiated but remain pluripotent), but rather, is broadly consistent with it and the subsequent literature that refined it. Overall, whilst I applaud the quality of analysis, the advancement of understanding is below a minimum threshold for 'new research' in my opinion.

Reviewer #3 (Remarks to the Author):

I appreciate the extremely thorough response from the authors. My points were all addressed. I am particularly happy to see the extended EpiLC differentiation experiment, which bolsters their model. I also thank the authors for clarifying the role of G9a, which was important, in my view. I enthusiastically endorse for publication.

Maxim Greenberg

Point-by-point response to the reviewers' comments

Reviewer #1 (Remarks to the Author):

Thank the authors for addressing all my questions. I recommend to have this MS published in NC.

We appreciate the reviewer's positive comments and recommendation for publication in NC.

Reviewer #2 (Remarks to the Author):

The authors have failed to adequately address my critical concern that there is highly limited novelty in the manuscript. The vast bulk of results (and conclusions) draws together the existing literature and paradigms; namely that DNA methylation becomes important for GGD silencing during differentiation in conjunction with SETDB1 and PRC1-E2F6-MGA-MAX, whereas naïve pluripotent cells, which are globally hypomethylated, preferentially rely on the latter systems for repression (see multiple references noted in original comments - not addressed). The authors argument that the unique novelty relies on identification of MGA independent of PCGF6 is highly nuanced (to the point it is not even articulated in the abstract) and also broadly established in any case (see recent PMID: 34117224 and prior work). Moreover this actually reflects a minor portion of the data; the great majority of the figures reinforce extant data and concepts.

Unfortunate that the reviewer 2 does not find our manuscript sufficiently novel. All of the previous studies the reviewer mentions were performed in primed ESCs and/or differentiated somatic cells with a high DNAm state, and therefore did not precisely describe the functions of MGA/PRC1.6/SETDB1 in the absence of DNAm in naïve ESCs (or pre-implantation embryos) that we have characterized here. As we discussed, we believe that the novelty of our manuscript lies in the combinatorial and systematic comparison of the disruption of multiple chromatin modifiers in the regulation of germline gene DNA methylation, covalent histone modifications and expression, coupled with the kinetics of establishment of these various chromatin marks during early embryonic development.

The excellent study specifically mentioned by the reviewer, namely Dahlet et al. (2021; PMID: 34117224), reported on the role of E2F6 in silencing of germline genes during embryonic development. Importantly, this study was published online (in Nature Communications coincidentally) on June 11th 2021. Our manuscript was submitted on April 6th, 2021 and sent out for peer review according to the editor on April 13th (it went live on Research Square on April 14th), nearly a month before the above-mentioned paper was published. The reviewer may not be aware of the "strength in numbers" policy of Nature Communications, found here- (<https://www.nature.com/articles/s41467-020-17817-x>) and quoted below:

"The robustness of science is best revealed when independent investigations of the same problem arrive at similar conclusions. At Nature Communications, we commit to disregard from our editorial evaluation any competing works that are published while a submission to our journal is under review or under revision by the authors."

This policy further states:

"In addition to our policy during review and revision, we will also adopt a more open stance when assessing new submissions whose results overlap considerably with a recently published paper. While each manuscript will still be judged on its own merits on a case-by-case basis, we will not

consider the presence of a recent publication reporting similar results as a sole reason for declining publication of the work being evaluated, when it is clear that the two studies were carried out independently.”

We have edited the manuscript text to make it clear, when citing Dahlet et al. (2021; PMID: 34117224), that this study was published after our manuscript was submitted. Based on the explanation provided above and the existence of this policy, we respectfully disagree with this reviewer’s point of view.

Finally, in naïve cells GGD are repressed but not silenced (expression is detected), thus full silencing as opposed to dampening does seem to be linked with acquisition of DNAm in differentiated cells in most cases (in collaboration with PRC/SETDB1/MGA, as established in several papers). To me the authors data does not seem to contrast with the central conclusions of PMID: 22949617 that DNAm is critical in differentiated tissues (EpiLC are not differentiated but remain pluripotent), but rather, is broadly consistent with it and the subsequent literature that refined it. Overall, whilst I applaud the quality of analysis, the advancement of understanding is below a minimum threshold for ‘new research’ in my opinion.

We show that transcriptional repression of GGD genes occurs in naïve ESCs (and very likely the pre-implantation embryo), in an MGA/PRC1.6 and SETDB1 dependent manner, independent of DNAmethylation. Furthermore, we show that a subset of GGD genes are repressed at least in part by the same machinery, independent of DNAmethylation, even in EpiLCs, from which germ cells can be derived. We believe that these observations do contrast with one of the central conclusions drawn by Hackett *et al.* (2012; PMID: 22949617). Quoting from the Abstract of this publication: “...we identified a core set of germline-specific genes that are dependent exclusively on promoter DNA methylation for initiation and maintenance of developmental silencing”.

We and others have conclusively show that these germline-specific genes are NOT dependent exclusively on promoter DNA methylation for initiation of transcriptional repression at earlier developmental stages than those analyzed by Hackett *et al.* and that the establishment of DNAmethylation in the promoter regions of several of these genes is at least in part dependent upon these alternative repressive pathways. While it is true that these genes are not fully “silenced” in naïve cells, this seems a semantic point, as clearly GGD expression levels are significantly higher in mutants of the factors we studied and the establishment of DNAmethylation at these genes likely depends on their adoption of a closed chromatin structure that depends on these alternative repressive pathways. DNAmethylation may be associated at late developmental stages (ie in some differentiated tissues) with silencing (using the reviewer’s definition), rather than incomplete repression of these germline genes, as a consequence of the absence of expression of transcription factors that promote their expression, which may be expressed at high levels in pluripotent and germ cells.

The authors then begin the DISCUSSION, with the following statement: “Our initial aim was to unambiguously identify genes that are directly and causally regulated by promoter DNA methylation”, and go on to claim, when referring to several of the germline genes in question, that: “Importantly, acquisition of DNA methylation at these promoters is both necessary for, and temporally coincides with, the initiation of transcriptional silencing at these loci. Indeed, silenced germline genes are preferentially associated with specialised promoter chromatin that is highly depleted of many repressive histone modifications, thus placing promoter CpG methylation as the predominant regulatory mechanism”. They go on to state that: “Our data supports a model whereby promoter DNA methylation is a primary mechanism for both initiating and maintaining silencing at a small number of germline-specific loci and is potentially the only epigenetic barrier to their ectopic expression.” And that: “...the absence of significant regulation by chromatin-based mechanisms at these genes makes them particularly sensitive to global demethylation events...”. Finally, they state

that: “Intriguingly, these gene promoters appear to contain a specialised chromatin domain that does not acquire repressive H3K27me3, H3K9me2, H3K9me3 or H4K20me3 histone modifications when silenced by DNA methylation. This later claim is based on analysis of NIH 3T3 cells, with no analyses, including of DNAm, at developmental stages preceding E6.5. This is not to say that this was not an important study, but rather that the conclusions drawn about the initiation of silencing of GGD genes were not based on any analyses of the histone marks present, including H3K27me3, H3K9me3 and H2Aub, in the promoter regions of these genes, either prior to or during the establishment of DNAm. Our study and others show that the transcriptional repression of these genes is clearly more complex than simply the presence of DNAm and we further show the timing of establishment of these alternative marks relative to DNAm in the developing embryo, an analysis that had not been done as comprehensively prior to our study, including that of Dahlet et al. Thus, we again respectfully disagree that the advancement of understanding of our study is below a minimum threshold for ‘new research’.

Reviewer #3 (Remarks to the Author):

I appreciate the extremely thorough response from the authors. My points were all addressed. I am particularly happy to see the extended EpiLC differentiation experiment, which bolsters their model. I also thank the authors for clarifying the role of G9a, which was important, in my view. I enthusiastically endorse for publication.

We appreciate the reviewer’s positive comments and strong endorsement of our manuscript for publication in NC.